# A unified theory of strong coupling Bose polarons:
# From repulsive polarons to non-Gaussian many-body bound states

Nader Mostaan,[1, 2, 3, *] Nathan Goldman,[3, †] and Fabian Grusdt[1, 2, ‡]

[1]*Department of Physics and Arnold Sommerfeld Center for Theoretical Physics (ASC),*
*Ludwig-Maximilians-Universität München, Theresienstr. 37, D-80333 München, Germany*
[2]*Munich Center for Quantum Science and Technology (MCQST), Schellingstr. 4, D-80799 München, Germany*
[3]*CENOLI, Université Libre de Bruxelles, CP 231, Campus Plaine, B-1050 Brussels, Belgium*
(Dated: June 16, 2023)

We address the Bose polaron problem of a mobile impurity interacting strongly with a host Bose-Einstein condensate (BEC) through a Feshbach resonance. On the repulsive side at strong couplings, theoretical approaches predict two distinct polaron branches corresponding to attractive and repulsive polarons, but it remains unclear how the two are related. This is partly due to the challenges resulting from a competition of strongly attractive (destabilizing) impurity-boson interactions with weakly repulsive (stabilizing) boson-boson interactions, whose interplay is difficult to describe with contemporary theoretical methods. Here we develop a powerful variational framework that combines Gaussian correlations among impurity-boson scattering states, including up to an infinite number of bosonic excitations, with exact non-Gaussian correlations among bosons occupying an impurity-boson bound state. This variational scheme enables a full treatment of strong nonlinearities arising in the Feshbach molecule on the repulsive side of the resonance. Within this framework, we demonstrate that the interplay of impurity-induced instability and stabilization by repulsive boson-boson interactions results in a discrete set of metastable many-body bound states at intermediate energies between the attractive and repulsive polaron branches. These states exhibit strong quantum statistical characteristics in the form of non-Gaussian quantum correlations, requiring non-perturbative beyond mean-field treatments for their characterization. Furthermore, these many-body bound states have sizable molecular spectral weights, accessible via molecular spectroscopy techniques. This work provides a unified theory of attractive and repulsive Bose polarons on the repulsive side of the Feshbach resonance.

## I. INTRODUCTION

Explaining the behavior of quantum materials through the notion of quasiparticles is a central paradigm in condensed matter physics. While many phases of matter, such as conventional superconductors and Fermi liquids, possess quasiparticle-like excitations [1–3], in some strongly correlated phases, such as strange metals, the excitation spectra defy quasiparticle-based descriptions [4–8]. Thus, studying the detailed mechanisms of quasiparticle formation and breakdown is of prime interest. An emblematic scenario for quasiparticle formation is the dressing of electrons in solid-state systems by lattice vibrations, giving rise to a quasiparticle termed *polaron*. Since its first formulation by Landau and Pekar [9], the polaron concept has been central to describing electron mobility in organic semiconductors [10–15], exciton transport in light-harvesting complexes [16–18], and phonon-based theories of high-temperature superconductivity [19–24]. The problem of characterization and description of polarons naturally falls in the broader context of *mobile quantum impurity problems*, where a single mobile impurity interacts with the elementary excitations of a many-body medium and gives rise to a quasiparticle

with renormalized properties.

Recent developments in the realization of synthetic quantum systems with increasing degrees of control and tunability resulted in an upsurge in research on mobile quantum impurity problems, both in fermionic [25–34] and bosonic [35–45] systems. In the latter case, the quasiparticle formed from an impurity resonantly coupled to a bosonic medium in a Bose-Einstein Condensate (BEC) phase is called *Bose polaron*. Numerous theoretical works have studied different properties of Bose polarons, including spectral response and quasiparticle properties [35, 36, 42, 44, 46–49], the implication of three-body correlations on the state of Bose polarons [50–54] and finite-temperature effects [55–57], to name a few. The powerful toolbox available in atomic gas settings has enabled the investigation of various aspects of Bose polaron physics, reaching impurity-medium interactions deep into the strong coupling regime. Contrary to its weak coupling counterpart, the strong coupling regime poses substantial challenges to both experiments and theory and comes with many aspects that, as we now review, are still poorly understood.

In particular, a unified theoretical framework is lacking that could describe the connection of repulsive and attractive polarons. The mainly employed theoretical methods so far either included an infinite number of weakly correlated excitations in the polaron cloud [49, 58, 59] or a highly restricted number of potentially strongly correlated excitations [35, 42, 43]. On the repul-

* [nader.mostaan@physik.uni-muenchen.de](mailto:nader.mostaan@physik.uni-muenchen.de)
† [nathan.goldman@ulb.be](mailto:nathan.goldman@ulb.be)
‡ [fabian.grusdt@physik.uni-muenchen.de](mailto:fabian.grusdt@physik.uni-muenchen.de)

arXiv:2305.00835v3 [cond-mat.quant-gas] 13 Jun 2023

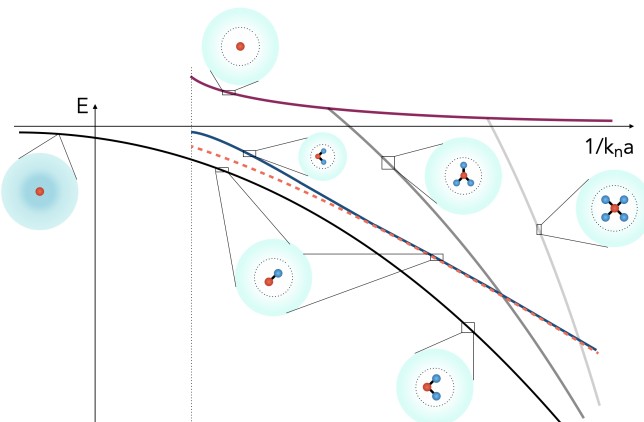

FIG. 1. Schematic illustration of the Bose polaron spectrum across an impurity-boson Feshbach resonance for repulsively interacting bosons. In the presence of inter-boson interactions, the attractive polaron persists to the repulsive side as a well-defined resonance, while other metastable many-body bound states appear in addition to the repulsive polaron. These many-body bound states emerge due to the competition of multiple impurity-boson binding and inter-boson repulsion. The structure of the main component of each many-body bound state is shown schematically.

sive side of the Feshbach resonance the former approaches do not include an attractive polaron branch. In contrast, the latter approaches predict that the attractive polaron continuously evolves into the molecular dimer state, energetically well below the metastable repulsive polaron, as the Feshbach resonance is crossed. We will argue in this article that neither of these scenarios is completely correct.

The peculiar nature of the Bose polaron problem at strong couplings becomes clearer by considering a static impurity interacting with an ideal, i.e. non-interacting BEC via an attractive potential. The strong coupling regime occurs when the impurity-boson potential admits a bound state with energy $-E_{\mathrm{B}} < 0$. In this regime, beyond a certain scattering length, a long-lived metastable polaronic state with energy $E_{\mathrm{RP}} > 0$ emerges, known as repulsive polaron, that involves the depletion of bosons close to the impurity from the polaron cloud. The repulsive polaron is unstable against the decay of bosons to the bound state. However, the number of decaying particles is not restricted for bosons, unlike fermions where Pauli blocking inhibits multiple occupations of the bound state. Thus, successive decay of bosons is energetically favorable, with a gain in energy per particle equal to the impurity-boson binding energy. In this sense, the spectrum of the system consists of an incoherent continuum of excitations on top of the repulsive polaron, together with a discrete set of bound states with energies $E_n = E_{\mathrm{RP}} - n E_{\mathrm{B}}$ for $n = 1, 2, 3, \cdots$, involving $n$ particles bound to the impurity.

This pathological behavior, first noted in Ref. [49] and discussed in further detail in [59], initiated active the-

oretical research to improve theoretical models that include the repulsive inter-boson interactions to counteract the impurity-induced instability of the ground state. In the simple model described above, including an effective inter-boson repulsion term $U n^2 / 2$ stabilizes the system. The ground state is then realized for $n^* = E_{\mathrm{B}} / U$ bosons, giving a finite ground state energy $E_{\mathrm{RP}} - E_{\mathrm{B}}^2 / 2U$ (see Fig. 1 for a schematic illustration). Thus, an effective repulsive interaction among bosons makes the model stable. In reality, this repulsion-induced stabilization is manifested via short-range repulsion of bosons close to the impurity, signifying the importance of short-range effects. Besides, an ideal theoretical description of strong coupling Bose polarons must involve an indefinite number of interacting bosons in the polaron cloud to properly capture the local correlations around the impurity while interpolating to long-length scales to account for the distortion of the condensate, rendering the problem theoretically challenging.

Recent theoretical works have analyzed the ground state energy by treating inter-boson interaction at the mean-field level [60–63], showing that the ground state energy remains finite in the thermodynamic limit. Exact Monte Carlo results [64] demonstrated that the theoretically employed classical field treatment accurately describes the polaron cloud of a static impurity supporting a bound state, when the impurity-boson interaction range $r_0$ is much larger than the inter-boson scattering length $a_{\mathrm{B}}$. Interestingly, in this framework, the polaron cloud contains infinitely many bosons with boson number $N$ growing sub-dimensional with system's volume $V$ (that is $N/V \to 0$ as $N, V \to \infty$). Nevertheless, the polaron ground state energy remains finite as the main contribution to the energy comes from the bosons localized around the impurity and not in the polaron tail.

In the opposite limit $r_0 \ll a_{\mathrm{B}}$, the standard one-parameter modeling of the low-energy scattering processes in three dimensions by the scattering length has to be amended by inclusion of short range details of the impurity-boson and boson-boson interaction potentials [63]. A non-local version of the Gross-Pitaevskii theory was proposed for the regime $r_0 \ll a_{\mathrm{B}}$ [65], including a method to account for finite range effects. Furthermore, truncated basis variational (TBV) methods exist that allow for the inclusion of impurity-boson correlations exactly up to a few particles [48, 50]. The TBV methods are especially suitable for cold atom realization of Bose polarons where the impurity-boson system is described by a two-channel model. In such models, another stabilization mechanism was identified [66–68] whereby the exchange of a closed-channel dimer effectively removes the impurity from the system and reduces the number of bound bosons to only a few, even in a non-interacting bosonic system. Although these TBV methods predict polaron energy accurately, their accuracy is limited by their few particle nature for observables such as quasiparticle residue that are sensitive to the particle number. The proper inclusion of an indefinite number of excita-

tions while at the same time accounting for finite-range impurity-boson and boson-boson interactions remains a central challenge in the development of an all-coupling theory of Bose polarons in a BEC.

Thus far, theoretical works on strong coupling Bose polarons have mainly focused on the independent characterization of the repulsive and attractive Bose polaron branches. In this work, we refine the understanding of strong coupling Bose polarons by addressing the physics of several metastable states that appear in this regime in the form of many-body bound states in addition to attractive and repulsive polarons. The existence of such states is already signaled in the simple stability argument laid out above, namely, that until repulsive interactions penalize bound state occupation beyond $n^*$ bosons, all the $n-$times occupations of the bound state with $n \leq n^*$ are energetically favorable. Such many-body bound states were studied before in the context of Rydberg [69] and ionic [70] impurities immersed in bosonic quantum gases, and for neutral impurities in two dimensions [71]. While for Bose polarons, such metastable bound states have been predicted before [49], the crucial effects of inter-boson repulsion have not been included so far.

To characterize these metastable states, we develop a variational principle that is able to accommodate the effects outlined above in a numerically efficient manner, and is accurate as long as the bound state is well separated from the other states in the bosonic one-particle spectrum. This variational principle builds upon a phenomenological model we formulate that enables to capture the essential correlations relevant for strong coupling Bose polarons. Although this variational scheme is suitable for generic impurity-condensate systems in arbitrary dimensionality, as a concrete example we focus on cold atom systems and characterize the metastable bound states emerging on the repulsive side of an impurity-boson Feshbach resonance.

Our variational approach enables us to unveil interesting properties of these states. For instance, the variational energy of these metastable bound states lie in between the attractive and repulsive polaron branches, and behave non-monotonously with particle number, resulting in level-crossings among the states (see Fig. 1). Moreover, the statistics of bosons bound to the impurity in these states exhibit strong quantum mechanical features, including non-Gaussian quantum correlations and interaction-induced anti-bunching. While the quantitative aspects of these effects depend on the particular setting considered, the underlying physical principles are general, and we expect such effects to occur in a broad class of impurity-BEC systems. Our results pave the way for investigating the implications of these metastable many-body bound states for Bose polaron physics at strong couplings.

Overall, our approach provides a unified theory of repulsive and attractive Bose polarons: we argue that the remnant of the attractive polaron branch on the repulsive side of the Feshbach resonance *coincides with* the lowest-lying multi-boson bound state around the metastable repulsive polaron. As the resonance is crossed the attractive polaron adiabatically evolves first into a molecular bound state with (approximately) one bound boson – as proposed in Ref. [35] – but then continues to adiabatically evolve into an (approximate) two-boson-plus-impurity bound state, and so on. Thereby, the stable attractive polaron on the repulsive side of the Feshbach resonance, along with additional metastable many-body bound states, is understood as a necessary and direct consequence of having a metastable repulsive-polaron saddle-point; i.e. the repuslive polaron cannot exist without its attractive counterpart.

The rest of the paper is organized as follows: in Sec. II, we outline the theoretical formalism and introduce our variational principle. In Sec. III, we apply our theoretical method to the special case of cold atomic Bose polarons, extract their energies and quantum correlated nature revealed by quantum statistics of bosons in the bound state, and discuss possible experimental detection of these states by molecular spectroscopy. In Sec. IV we compare the variational scheme presented here to existing methods and discuss its merits and limitations. We conclude in Sec. V and draw several future directions.

## II. THEORETICAL FORMALISM

### A. Model

We consider a mobile impurity of mass $M$ coupled to a bosonic medium, consisting of particles of mass $m$ in a condensed phase with density $n_0$ in three dimensions. The boson-boson and impurity-boson interactions are modeled by single-channel central potentials $U_{\mathrm{BB}}(\mathbf{x})$ and $V_{\mathrm{IB}}(\mathbf{x})$, respectively. The impurity is described by its position and momentum operators $\hat{\mathbf{X}}$ and $\hat{\mathbf{P}} = -i\hbar\nabla_{\hat{\mathbf{X}}}$, and the bosonic environment by the field operators $\hat{\phi}_{\mathbf{x}}$ and $\hat{\phi}^{\dagger}_{\mathbf{x}}$ satisfying bosonic commutation relations $[\hat{\phi}_{\mathbf{x}}, \hat{\phi}^{\dagger}_{\mathbf{x'}}] = \delta^{(3)}(\mathbf{x} - \mathbf{x'})$, $[\hat{\phi}_{\mathbf{x}}, \hat{\phi}_{\mathbf{x'}}] = [\hat{\phi}^{\dagger}_{\mathbf{x}}, \hat{\phi}^{\dagger}_{\mathbf{x'}}] = 0$. It is convenient to treat the condensed system in a grand-canonical ensemble by introducing a chemical potential $\mu$ fixing the condensate's mean particle number.

The total Hamiltonian $\hat{H}_{\mathrm{tot}}$ describing the system takes the form

$$\hat{H}_{\mathrm{tot}} = \hat{\mathbf{P}}^2/2M + \int_{\mathbf{x}} V_{\mathrm{IB}}(\mathbf{x} - \hat{\mathbf{X}})\, \hat{\phi}^{\dagger}_{\mathbf{x}}\hat{\phi}_{\mathbf{x}} + \hat{H}_{\mathrm{B}}\,, \qquad (1)$$

with $\int_{\mathbf{x}} \equiv \int d^3x$. It consists of the impurity kinetic energy, impurity-boson interaction, and the bosonic Hamiltonian $\hat{H}_{\mathrm{B}}$, given by

$$\begin{aligned} \hat{H}_{\mathrm{B}} = &\int_{\mathbf{x}} \hat{\phi}^{\dagger}_{\mathbf{x}}\big( -\hbar^2\nabla^2/2m - \mu\big)\hat{\phi}_{\mathbf{x}} \\ &+ \frac{1}{2}\int_{\mathbf{x},\mathbf{x'}} U_{\mathrm{BB}}(\mathbf{x} - \mathbf{x'})\, \hat{\phi}^{\dagger}_{\mathbf{x}}\hat{\phi}^{\dagger}_{\mathbf{x'}}\hat{\phi}_{\mathbf{x'}}\hat{\phi}_{\mathbf{x}}\,. \end{aligned} \qquad (2)$$

The problem is further simplified by transforming to the frame co-moving with the impurity. This is achieved through the *Lee-Low-Pines* transformation [44] $\hat{U}_{\text{LLP}} = \exp\left(i/\hbar\,\hat{\mathbf{X}}\cdot\hat{\mathbf{P}}_{\text{bath}}\right)$, where $\hat{\mathbf{P}}_{\text{bath}} = \int_{\mathbf{x}}\hat{\phi}^{\dagger}_{\mathbf{x}}(-i\hbar\nabla_{\mathbf{x}})\hat{\phi}_{\mathbf{x}}$ is the total momentum operator of the bath. Under $\hat{U}_{\text{LLP}}$, a state $|\Psi(\mathbf{K}_0)\rangle$ with well-defined total momentum $\mathbf{K}_0$ transforms to

$$|\Psi(\mathbf{K}_0)\rangle_{\text{LLP}} = \hat{U}_{\text{LLP}}|\Psi(\mathbf{K}_0)\rangle = |\mathbf{K}_0\rangle_{\text{imp}}\otimes|\Psi_{\mathbf{K}_0}\rangle_{\text{bath}}\,,\tag{3}$$

which enables restricting the total Hilbert space to the sector with well-defined impurity momentum $\mathbf{K}_0$. The transformed total Hamiltonian under $\hat{U}_{\text{LLP}}$ reads

$$\begin{aligned}\hat{H}_{\text{LLP}} &= \frac{\hbar^2}{2M}\mathbf{K}_0^2 - \frac{\hbar}{M}\mathbf{K}_0\cdot\hat{\mathbf{P}}_{\text{bath}} + \frac{:\hat{\mathbf{P}}_{\text{bath}}^2:}{2M}\\ &+ \int_{\mathbf{x}}V_{\text{IB}}(\mathbf{x})\,\hat{\phi}^{\dagger}_{\mathbf{x}}\hat{\phi}_{\mathbf{x}}\\ &+ \int_{\mathbf{x}}\hat{\phi}^{\dagger}_{\mathbf{x}}\big(-\hbar^2\nabla^2/2m_{\text{red}} - \mu\big)\hat{\phi}_{\mathbf{x}}\\ &+ \frac{1}{2}\int_{\mathbf{x},\mathbf{x}'}U_{\text{BB}}(\mathbf{x}-\mathbf{x}')\,\hat{\phi}^{\dagger}_{\mathbf{x}}\hat{\phi}^{\dagger}_{\mathbf{x}'}\hat{\phi}_{\mathbf{x}'}\hat{\phi}_{\mathbf{x}}\,,\end{aligned}\tag{4}$$

where $\mathbf{K}_0$ is the total momentum of the system, $m_{\text{red}}^{-1} = m^{-1} + M^{-1}$ is the impurity-boson reduced mass, and $:\cdots:$ denotes normal ordering of field operators. Eq. 4 is obtained using $\hat{U}_{\text{LLP}}^{\dagger}\hat{\mathbf{P}}\,\hat{U}_{\text{LLP}} = \hat{\mathbf{P}} - \hat{\mathbf{P}}_{\text{bath}}$ and the replacement $\hat{\mathbf{P}}\to\mathbf{K}_0$ on the restricted Hilbert space. In the rest of the paper we focus on the case $\mathbf{K}_0 = 0$, which corresponds to the overall ground state.

After introducing the model Hamiltonian, it is instructive to adopt a path integral formalism to study strong coupling Bose polarons. Path integral formulation is able to represent Bose polaron models in dense and dilute media and capture crucial strong coupling effects such as impurity-induced instability and condensate deformation. The free energy $F$ of the system in path integral representation takes the following form

$$e^{iF/\hbar} = \int\mathcal{D}[\varphi^*,\varphi]\,e^{i\mathcal{S}[\varphi^*,\varphi]/\hbar}\,,\tag{5}$$

where $\mathcal{S}[\varphi^*,\varphi]$ is the action in terms of the classical fields $\varphi^*$ and $\varphi$, written as

$$\mathcal{S}[\varphi^*,\varphi] = \int d^{3+1}x\left(\varphi^*\,i\hbar\partial_t\varphi - H_{\text{LLP}}[\varphi^*,\varphi]\right).\tag{6}$$

It is standard to treat $F$ within a saddle point approximation, that involves finding the saddle points of $\mathcal{S}[\varphi^*,\varphi]$.

Crucially, the saddle point analysis of the action reveals the existence of repulsive and attractive polarons on the repulsive side of the Feshbach resonance as the unstable, respectively, stable saddle points of the action. It is a key messages of our work to underline the necessity of going beyond the saddle point approximation to study the physics of metastable many-body bound states, as

those states emerge due to the strong modification of the energy landscape around the repulsive polaron by inter-boson interactions. Nevertheless, as a starting point of the theoretical construction it is necessary to outline a detailed picture of the saddle point structure of the model. This is the topic of the next subsection.

### B. Saddle point analysis

#### 1. Mean-field decoupling of $\hat{H}_{\text{LLP}}$

To obtain the saddle point solutions and analyze the associated energy landscape, it is instructive to perform a mean-field decoupling of the Hamiltonian. To this end, we separate $\hat{\phi}_{\mathbf{x}}$ into a classical component $\varphi_{\mathbf{x}}$ representing the condensate, and quantum fluctuations $\delta\hat{\phi}_{\mathbf{x}}$, i.e. $\hat{\phi}_{\mathbf{x}} = \varphi_{\mathbf{x}} + \delta\hat{\phi}_{\mathbf{x}}$. For notational convenience, we introduce the Nambu vector $\delta\hat{\Psi}$ with coordinate representation $\delta\hat{\Psi}_{\mathbf{x}} = (\delta\hat{\phi}_{\mathbf{x}},\delta\hat{\phi}^{\dagger}_{\mathbf{x}})^T$.

Within the mean-field theory, the elementary excitations of the system are modeled by weakly interacting quasiparticles with Bogoliubov-type field operators $\hat{B}_{\mathbf{x}} = (\hat{\beta}_{\mathbf{x}},\hat{\beta}^{\dagger}_{\mathbf{x}})^T$ related to $\delta\hat{\Psi}$ through the canonical transformation $\delta\hat{\Psi}_{\mathbf{x}} = \int_{\mathbf{y}}S_{\mathbf{xy}}\hat{B}_{\mathbf{y}}$, where $S_{\mathbf{xy}}$ are $2\times 2$ matrices. Note that both the classical component $\varphi_{\mathbf{x}}$ as well as the Bogoliubov modes $\hat{B}_{\mathbf{x}}$ should be calculated in the presence of the impurity in the Lee-Low-Pines frame, as explained below.

Correspondingly, the vacuum state of elementary excitations $|\text{GS}\rangle$, defined by $\hat{\beta}_{\mathbf{x}}|\text{GS}\rangle = 0$, is connected to the bosonic vacuum $|\emptyset\rangle$ by $|\text{GS}\rangle = \hat{\mathcal{S}}|\emptyset\rangle$ where

$$\hat{\mathcal{S}} = \exp\left(\frac{i}{2}\delta\hat{\Psi}^{\dagger}\,\Xi\,\delta\hat{\Psi}\right),\tag{7}$$

is a bosonic squeezing operator. In Eq. 7, $\Xi$ is a Hermitian matrix related to $S$ by $S = \exp(i\Sigma_z\Xi)$ with $\Sigma_z = \sigma_z\delta^{(3)}(\mathbf{x}-\mathbf{x}')$ and $\sigma_z$ the Pauli-$z$ operator. For shorthand notation, matrix multiplication implies integration over spatial coordinates and summation over Nambu components. To fulfill the bosonic commutation relations for $\hat{\beta}_{\mathbf{x}}$ and $\hat{\beta}^{\dagger}_{\mathbf{x}}$, $S$ must be a symplectic matrix satisfying $S^{\dagger}\Sigma_z S = \Sigma_z$.

By means of Wick's theorem, $\hat{H}_{\text{LLP}}$ takes the form (see Appendix A)

$$\begin{aligned}\hat{H}_{\text{LLP}} &= E[\Phi,\Gamma] + \left(\delta\hat{\Psi}^{\dagger}\cdot\zeta[\Phi,\Gamma] + h.c.\right)\\ &+ \frac{1}{2}:\delta\hat{\Psi}^{\dagger}\mathcal{H}_{\text{MF}}[\Phi,\Gamma]\delta\hat{\Psi}: + \hat{H}_3 + \hat{H}_4\,.\end{aligned}\tag{8}$$

Here, $\Phi_{\mathbf{x}} = (\varphi_{\mathbf{x}},\varphi^*_{\mathbf{x}})^T$, the covariance matrix $\Gamma$ is defined by $2\Gamma = \langle\text{GS}|\{\delta\hat{\Psi},\delta\hat{\Psi}^{\dagger}\}|\text{GS}\rangle - \mathbb{I}$ and can be expressed in terms of $S$ by $2\Gamma + \mathbb{I} = SS^{\dagger}$, $\mathbb{I}$ is the identity matrix and $:\cdots:$ denotes normal ordering with respect to $\hat{\beta}_{\mathbf{x}}$ and $\hat{\beta}^{\dagger}_{\mathbf{x}}$. Furthermore, $\mathcal{H}_{\text{MF}}[\Phi,\Gamma]$ is the mean-field

Hamiltonian, $\hat{H}_3$ and $\hat{H}_4$ are the cubic and quartic Hamiltonians in the field operators, respectively, and $\zeta[\Phi, \Gamma]$ is defined in Appendix A.

In standard mean-field theory, beyond quadratic terms are neglected, while $\Phi_0$ and $S_0$ are found that correspond to the saddle point solution $\zeta[\Phi_0, \Gamma_0] = 0$ and diagonalize the mean-field Hamiltonian as $S_0^\dagger \mathcal{H}_{\mathrm{MF}}[\Phi_0, \Gamma_0] S_0 = \mathbb{I}_2 \otimes D$, with $\mathbb{I}_2$ the $2 \times 2$ identity matrix and $D$ a diagonal matrix. The condition $2\Gamma_0 + \mathbb{I} = S_0 S_0^\dagger$ and the dependence of $\mathcal{H}_{\mathrm{MF}}$ on $\Gamma_0$ require that $S_0$ be obtained self-consistently. The resulting normal modes $\hat{B}_0 = S_0^{-1} \delta \hat{\Psi}$ are the well-known Bogoliubov modes.

In the following, we analyze the quadratic terms in $\hat{H}_{\mathrm{LLP}}$ from a mean-field viewpoint. However, as we elucidate later, it is crucial to retain the higher-order terms $\hat{H}_3$ and $\hat{H}_4$ to describe essential strong coupling effects such as the non-Gaussian correlations of Bose polaron many-body bound states at strong couplings.

### 2. Saddle point structure

Next, we analyze the saddle point and normal mode structure of the quadratic part of $\hat{H}_{\mathrm{LLP}}$ across an impurity-boson scattering resonance. On the attractive side ($a < 0$, with $a$ the impurity-boson scattering length), the saddle point condition is equivalent to the Gross-Pitaevskii equation and admits a single solution $\Phi_{\mathrm{att}}$ that is the attractive polaron (dashed green line in Fig. 2). The static and dynamic properties of the attractive polaron obtained within Gross-Pitaevskii were investigated in [61–63, 65], and the predictions for cold atom settings are in excellent agreement with the experiments. Furthermore, the attractive polaron is a stable saddle point solution, meaning that all the corresponding fluctuation modes have positive energy, or equivalently, $\mathcal{H}_{\mathrm{MF}}[\Phi_0, \Gamma_0]$ is positive-definite (see Fig. 2(b), panel (1)).

The attractive polaron solution extends to the repulsive side ($a > 0$) and remains a stable saddle point. Nevertheless, for the mean-field Hamiltonian $\mathcal{H}_{\mathrm{MF}}[\Phi, \Gamma]$, there exists a dynamical instability window of impurity-boson interaction strength, where an unstable phase quadrature of a Bogoliubov mode emerges [58] (see Fig. 2(b), panel (2)).

Beyond the dynamical instability, another saddle point solution $\Phi_{\mathrm{rep}}$ emerges that is the repulsive polaron. The repulsive polaron saddle point is unstable, as a single Bogoliubov mode with negative energy exists in the spectrum of $\mathcal{H}_{\mathrm{MF}}[\Phi_{\mathrm{rep}}, \Gamma_{\mathrm{rep}}]$. The existence of this unstable mode is traced back to the bound state of the impurity-boson potential, therefore with a slight abuse of terminology, we call it "the bound state" or "dimer" as well (see Appendix B for further discussion on the bound Bogoliubov mode and its relation to the impurity-boson bound state). Analogously, we call the extended modes with positive energy "scattering Bogoliubov modes" or "scattering states". In fact, when $V_{\mathrm{IB}}$ admits $\nu$ bound states, there exists $2\nu + 1$ solution to the Gross-Pitaevskii equa-

tion; see Refs. [61, 72]. We leave the study of the third solution to the Gross-Pitaevskii equation for future research.

In a mean-field treatment of the Bose polaron without including inter-boson interactions [58], the presence of the unstable mode implies that the system can decrease its energy by filling the bound state with bosons, resulting in the many-body ground state energy $E_{\mathrm{GS}} = -\infty$. This pathological behavior signifies the need for a nonperturbative beyond mean-field treatment of the Bose polaron by the full Hamiltonian in Eq. 8, i.e. including the cubic and quartic terms.

While an exact non-perturbative solution for the spectrum of $\hat{H}_{\mathrm{LLP}}$ is infeasible due to the strongly correlated nature of the problem, one can capture the essential correlations using a phenomenological model, while rendering a stable state analysis of the problem possible. The formulation of this phenomenological model is one of the main results of our work. In the following we introduce the effective model we devise for investigating Bose polarons at strong impurity-boson interactions.

### C. Effective Model and variational principle

The first step to obtain the effective model is to harness the large seperation of energy scales between the scattering states and the bound state of the mean-field Hamiltonian at strong couplings. This large separation of energy and length scales enables to treat the bound state separately from the rest of the modes. Formally, this separation is achieved by splitting the bosonic annihilation operator into two parts, $\hat{\phi}_{\mathbf{x}} = \hat{\phi}_{\mathbf{x}}^{(\mathrm{B})} + \hat{\phi}_{\mathbf{x}}^{(\mathrm{sc})}$. Here, $\hat{\phi}_{\mathbf{x}}^{(\mathrm{B})} = (u_{\mathrm{B},\mathbf{x}} \hat{b} + v_{\mathrm{B},\mathbf{x}} \hat{b}^\dagger)$, $u_{\mathrm{B},\mathbf{x}}$ and $v_{\mathrm{B},\mathbf{x}}$ are the real space form of Bogoliubov factors associated to the bound Bogoliubov mode, $\hat{b}$ is its annihilation operator, and $\hat{\phi}_{\mathbf{x}}^{(\mathrm{sc})} = \hat{\phi}_{\mathbf{x}} - \hat{\phi}_{\mathbf{x}}^{(\mathrm{B})}$ only consists of scattering Bogoliubov modes. We deploy this mode separation to recast the Hamiltonian $\hat{H}_{\mathrm{LLP}}$ to a form that is more appropriate for our variational treatment later on. With this mode separation, the Hamiltonian $\hat{H}_{\mathrm{LLP}}$ of Eq. 8 takes the following form,

$$\hat{H}_{\mathrm{LLP}} = \sum_{\substack{n, m \\ n+m=4}} \hat{b}^{\dagger n} \hat{b}^m \, \hat{H}_{n,m}[\hat{\phi}_{\mathbf{x}}^{(\mathrm{sc})\dagger}, \hat{\phi}_{\mathbf{x}}^{(\mathrm{sc})}], \qquad (9)$$

where $\hat{H}_{n,m}[\hat{\phi}_{\mathbf{x}}^{(\mathrm{sc})\dagger}, \hat{\phi}_{\mathbf{x}}^{(\mathrm{sc})}]$ terms only act on the scattering Bogoliubov modes, and $n, m$ denote powers of the bound Bogoliubov mode operators. Note that to obtain the form in Eq. 9, the mean-field decoupling of $\hat{H}_{\mathrm{LLP}}$ has to be performed over the repulsive polaron saddle point, with the corresponding condensate field $\Phi_{\mathrm{rep}}$ and covariance matrix $\Gamma_{\mathrm{rep}}$. This is again because the bound Bogoliubov mode is a well-defined unstable mode of the repulsive polaron saddle point.

We now introduce the structure of variational states to model the metastable many-body bound states. First,

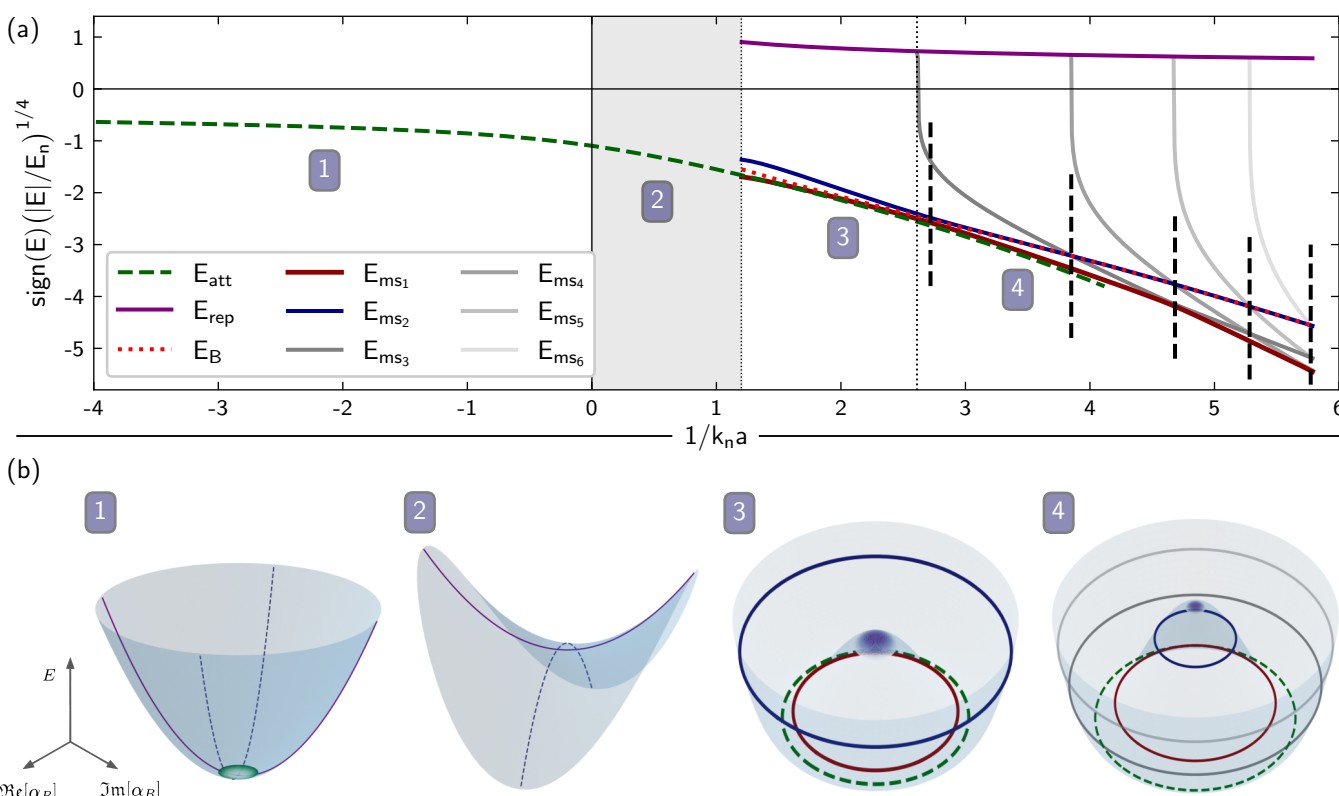

FIG. 2. (a) Energy of polaron states, including attractive and repulsive polaron, and metastable states $ms_1$ to $ms_6$ (see text), across an impurity-boson Feshbach resonance. On the attractive side ($a < 0$), an impurity resonance exists corresponding to the attractive polaron branch (green dashed line), which extends to the repulsive side and remains the well-defined stable saddle point across the resonance. On the repulsive side, the repulsive polaron branch emerges as the unstable saddle point solution with a bound state, as well as two many-body bound states $ms_1$ and $ms_2$ (red and blue solid lines). The red dotted line indicates the bare dimer energy. Beyond a critical scattering length (denoted by a vertical black dotted line), further metastable many-body bound states $ms_3$ to $ms_6$ emerge in the spectrum (grey shaded solid lines). Note that the normalized energy is rescaled to show all bound states compactly. The grey-shaded region (2) on the repulsive side is bounded by $1/k_n a \simeq 1.2$ where $\mu/\varepsilon_B \simeq 9 \times 10^{-3}$, providing a conservative bound for the validity of our theory. (b) The energy landscape over the phase space of the bound Bogoliubov mode, around the saddle points corresponding to different regions in (a). The real and imaginary parts of the coherent state variable $\alpha_B$ serve as coordinates for the phase space of the bound Bogoliubov mode. In (1), the attractive polaron (green shaded point) is a stable saddle point, with all the fluctuation modes having positive energy. Within region (2), a dynamical instability occurs as a precursor to the formation of the repulsive polaron, signified by a single unstable phase mode with a corresponding stable amplitude mode. In (3), the repulsive polaron (purple shaded dot) is a saddle point solution with a single unstable Bogoliubov mode. The energy and particle number of many-body bound states in (a) are depicted qualitatively on the energy surfaces. The radius of each circle denotes the mean bound state occupation number, while its position on the surface denotes the energy of the state. Repulsive inter-boson interaction increases the energy of the many-body bound state with a higher particle number. By increasing $1/k_n a$, further many-body bound-states enter the atom-dimer continuum (grey shaded solid lines). Increasing the binding energy increases the number of bound bosons in the lowest many-body bound state. The vertical black dashed lines mark the level crossings between many-body bound states.

we note that an arbitrary eigenstate of $\hat{H}_{\text{LLP}}$ Eq. 9 with energy $E$ can be decomposed into $|\psi_E\rangle = \sum_n a_{n,E} |n\rangle_B \otimes |\psi_{n,E}\rangle_{\text{sc}}$, where $a_{n,E}$ for $n = 0, 1, 2, \cdots$ are coefficients, $|n\rangle_B = \hat{b}^{\dagger n}/\sqrt{n!} |\text{GS}\rangle$ is the Fock state of the bound Bogoliubov mode, and $|\psi_{n,E}\rangle_{\text{sc}}$ is a corresponding many-body state of the scattering Bogoliubov modes.

Using the separation of time scales over which the bound and scattering Bogoliubov modes evolve, we require the variational states $|\psi_{(\text{var})}\rangle$ approximating $|\psi_E\rangle$ to be separable in the Hilbert space of the bound and

scattering Bogoliubov modes as

$$|\psi_{(\text{var})}\rangle = |\psi_{(B)}\rangle_B \otimes |\psi_{(\text{sc})}\rangle_{\text{sc}}, \qquad (10)$$

where the additional subscripts "B" and "sc" refer to the Hilbert spaces of the bound and scattering Bogoliubov modes, respectively, and we drop them hereafter. This approximation is in the spirit of the Born-Oppenheimer approximation [73] used frequently in quantum chemistry to determine the electronic structure of a molecule, by using the separation of energy scales between the fast

and slow degrees of freedom. One then assumes that fast degrees of freedom adiabatically follow the dynamics of the slow degrees of freedom. In the present context, the bound and scattering Bogoliubov modes constitute the fast and slow degrees of freedom, respectively.

Following the same reasoning, we identify $\left|\psi_{(\mathrm{B})}\right\rangle$ as the eigenstate of the effective Hamiltonian

$$
\begin{aligned}
\hat{H}_{\mathrm{eff},\mathrm{B}} &= \left\langle\psi_{(\mathrm{sc})}\right| \hat{H}_{\mathrm{LLP}} \left|\psi_{(\mathrm{sc})}\right\rangle \\
&= \sum_{\substack{n,m \\ n+m=4}} \left\langle\psi_{(\mathrm{sc})}\right| \hat{H}_{n,m}[\hat{\phi}_{\mathbf{x}}^{(\mathrm{sc})\dagger}, \hat{\phi}_{\mathbf{x}}^{(\mathrm{sc})}] \left|\psi_{(\mathrm{sc})}\right\rangle \, \hat{b}^{\dagger n}\hat{b}^{m} \,,
\end{aligned}
$$
(11)

while the effective Hamiltonian for scattering Bogoliubov modes reads

$$
\begin{aligned}
\hat{H}_{\mathrm{eff},\mathrm{sc}} &= \left\langle\psi_{(\mathrm{B})}\right| \hat{H}_{\mathrm{LLP}} \left|\psi_{(\mathrm{B})}\right\rangle \\
&= \sum_{\substack{n,m \\ n+m=4}} \left\langle\psi_{(\mathrm{B})}\right| \hat{b}^{\dagger n}\hat{b}^{m} \left|\psi_{(\mathrm{B})}\right\rangle \, \hat{H}_{n,m}[\hat{\phi}_{\mathbf{x}}^{(\mathrm{sc})\dagger}, \hat{\phi}_{\mathbf{x}}^{(\mathrm{sc})}] \,.
\end{aligned}
$$
(12)

To determine the variational structure of $\left|\psi_{(\mathrm{B})}\right\rangle$ and $\left|\psi_{(\mathrm{sc})}\right\rangle$, we take $\left|\psi_{(\mathrm{B})}\right\rangle$ to be an unrestricted superposition of Fock states $|n\rangle_{\mathrm{B}}$ as $\left|\psi_{(\mathrm{B})}\right\rangle = \sum_n \psi_n |n\rangle_{\mathrm{B}}$, while we take $\left|\psi_{(\mathrm{sc})}\right\rangle$ to be a coherent state

$$
|\alpha_{\mathbf{x}}\rangle = \exp\left( \int_{\mathbf{x}} \alpha_{\mathbf{x}}\, \delta\hat{\phi}_{\mathbf{x}}^{\dagger} - h.c. \right) |\Phi_{\mathrm{rep}}\rangle \,,
$$
(13)

where $\alpha_{\mathbf{x}}$ is the real space profile of the coherent cloud of bosons occupying the scattering Bogoliubov modes. We then obtain the complete form of the variational state as

$$
\left|\psi_{(\mathrm{var})}[\psi_n, \alpha_{\mathbf{x}}]\right\rangle = \left( \sum_n \psi_n |n\rangle_{\mathrm{B}} \right) \otimes |\alpha_{\mathbf{x}}\rangle \,.
$$
(14)

The Hamiltonian $\hat{H}_{\mathrm{LLP}}$ displayed as in Eq. 9, together with the variational states presented in Eq. 14, constitute the basis of our variational principle. The variational parameters $\psi_n$, $\alpha_{\mathbf{x}}$ and $\alpha_{\mathbf{x}}^*$ are then determined by optimizing the energy functional

$$
\begin{aligned}
H[\psi_n^*, &\psi_n, \alpha_{\mathbf{x}}^*, \alpha_{\mathbf{x}}] \\
&= \left\langle\psi_{(\mathrm{var})}[\psi_n, \alpha_{\mathbf{x}}]\right| \hat{H}_{\mathrm{LLP}} \left|\psi_{(\mathrm{var})}[\psi_n, \alpha_{\mathbf{x}}]\right\rangle \,,
\end{aligned}
$$
(15)

with respect to $\psi_n$ and $\alpha_{\mathbf{x}}$ subject to the conditions

$$
\left\langle\psi_{(\mathrm{var})}[\psi_n, \alpha_{\mathbf{x}}] \big| \psi_{(\mathrm{var})}[\psi_n, \alpha_{\mathbf{x}}]\right\rangle = 1 \,,
$$
(16)

$$
\int_{\mathbf{x}} \left( u_{\mathrm{B},\mathbf{x}}^* \alpha_{\mathbf{x}} - v_{\mathrm{B},\mathbf{x}} \alpha_{\mathbf{x}}^* \right) = 0 \,.
$$
(17)

The condition in Eq. 16 is the normalization of the variational wavefunction, while the condition in Eq. 17 results from the requirement that $|\alpha_{\mathbf{x}}\rangle$ consists of the scattering

Bogoliubov modes only, thus $\hat{b}\,|\alpha_{\mathbf{x}}\rangle = 0$. Note that the parameters $u_{\mathrm{B},\mathbf{x}}$, $v_{\mathrm{B},\mathbf{x}}$ are determined by the saddle-point solution of the repulsive polaron.

Some comments on the variational scheme presented above are in order. First, note that $\left|\psi_{(\mathrm{B})}\right\rangle$ is a many-body state composed of a superposition of Fock states of the bound Bogoliubov mode, hence the name "many-body bound state"; thus, it takes into account the quantum correlations of bound Bogoliubov excitations *exactly*, without restricting the number of excitations. Second, the assumption of separability of the eigenstates between bound and scattering Bogoliubov modes is justified as long as the energy of $\left|\psi_{(\mathrm{B})}\right\rangle$ remains well separated from the energy of the scattering Bogoliubov modes.

To explain the intuitive meaning of this second condition, we again resort to the simple model presented in the introduction, and note that all states with $n$-times occupation of the bound state where $n^* \leq n < 2n^*$ have energy less than the repulsive polaron. If the energy difference of the $\lfloor 2n^* \rfloor$ state (with $\lfloor n \rfloor$ the integer part of $n$) to the repulsive polaron is comparable to the typical energy of phonon excitations (which is of the order of the BEC chemical potential $\mu$), then a boson added to the bound state to construct the $\lfloor 2n^* \rfloor$ state from the $\lfloor 2n^* \rfloor - 1$ state would also have a comparable occupation of the scattering states. Requiring that $|E_{\lfloor 2n^* \rfloor}|$ be much larger than $\mu$, leads to $\mu/|\varepsilon_{\mathrm{B}} - U/2| \ll 1$. Applying the same argument to the effective model introduced here leads to the condition

$$
\mu \ll \left| H_{22} \lfloor 1 + \varepsilon_{\mathrm{B}}/H_{22} \rfloor \left( \varepsilon_{\mathrm{B}}/H_{22} - \lfloor \varepsilon_{\mathrm{B}}/H_{22} \rfloor \right) \right| \,, \quad (18)
$$

with $H_{22} = 1/2 \int_{\mathbf{x},\mathbf{x}'} U_{\mathrm{BB}}(\mathbf{x} - \mathbf{x}') |u_{\mathrm{B},\mathbf{x}}|^2 |u_{\mathrm{B},\mathbf{x}'}|^2$.

Third, regarding the assumption of coherent state occupation of scattering Bogoliubov modes, note that the bosons occupying the bound state are localized around the impurity. Thus they screen the impurity potential for the rest of the condensed bosons. This screening results in a modification of the condensate field that leads to the excitation of scattering Bogoliubov modes of the unperturbed condensate. This condensate distortion effect is captured by the coherent field $\alpha_{\mathbf{x}}$. In principle, an exact many-body wavefunction for the Bose polaron includes higher-order correlations and entanglement among the excited scattering Bogoliubov modes that goes beyond the uncorrelated coherent state. Nevertheless, for heavy impurities, the scattering Bogoliubov modes are now weakly interacting and delocalized, so the entanglement among these modes caused by their interactions - either mediated by the impurity or from higher-order processes - plays a negligible role. Thus, modeling the excitation of scattering Bogoliubov modes by a coherent state $|\alpha_{\mathbf{x}}\rangle$ is justified.

A final remark concerns the influence of three-body correlations on the spectrum of the system. Our analysis ignores the more complicated three-body correlations underlying Efimov states [51–53, 74]. This is fully justified for heavy impurities where the size of excited Efimov clusters is much larger than many-body bound

states considered here. For lighter impurities, the few-body bound states we describe are expected to decay into deeply bound Efimov states but we leave a detailed analysis of their influence to future research.

In the following, we apply our theory to a relevant experimental cold atoms setting and discuss some of the main features of the resulting many-body bound states on the repulsive side of the Feshbach resonance. As a key result, we reveal non-Gaussian quantum mechanical correlations in the bound state occupation statistics of these states.

## III. RESULTS

Here we consider a Bose polaron setting comprised of impurity $^{40}$K atoms immersed in a BEC of $^{87}$Rb atoms with condensate density $n_0 = 1.8 \times 10^{14}\,\mathrm{cm}^{-3}$ and inter-boson scattering length $a_\mathrm{B} = 100\,a_0$ with $a_0 = 0.529\,\text{Å}$ the Bohr radius [36]. The natural length and energy units are then the inverse Fermi momentum $k_n = (6\pi^2 n_0)^{1/3}$ and energy $E_n = \hbar^2 k_n^2/2m_\mathrm{B}$, respectively. The impurity-boson potential is modeled by a squarewell of the form $V_\mathrm{IB}(\mathbf{r}) = V_0\,\Theta(r_0 - r)$ where $r = |\mathbf{r}|$ and $r_0$ is the potential range tuned properly to retrieve the impurity-boson effective range. The boson-boson scattering potential can be modeled by a zero-range contact interaction $U_\mathrm{BB}(\mathbf{x}) = U_0\,\delta(\mathbf{x})$ compatible with the Born approximation. Note that the major effect of any finite boson-boson interaction range would appear in the interaction of bound Bogoliubov modes, while the bound-scattering and scattering-scattering mode interactions are still well modeled by contact boson-boson interactions. The latter is due to the fact that only low energy scattering Bogoliubov modes with momenta of the order of $1/\xi_\mathrm{red}$ are involved, with $\xi_\mathrm{red}^2 = \hbar^2/(2m_\mathrm{red}n_0 U_0)$ the modified BEC healing length, which is much larger than the boson-boson interaction range. Thus, we expect the effect of non-zero boson-boson interaction to be quantitative and only result in marginal changes in the interaction strength of bound Bogoliubov modes.

Having described the system, we now use the variational principle explained before to obtain the relevant stable-state solutions across the impurity-boson scattering resonance. To this end, we apply the construction presented earlier step-by-step. Furthermore, at each step we carry out suitable approximations that are applicable to the problem considered here and illustrate the essential physics in a more transparent manner.

The first step is to find the repulsive polaron saddle-point solution by the procedure outlined in Sec. II B 1. To find $\Phi_\mathrm{rep}$ and $S_\mathrm{rep}$, we begin by an initial guess $S_\mathrm{rep,0} = \mathbb{I}$, and solve $\zeta[\Phi_\mathrm{rep,0}, \mathbb{I}] = 0$. The resulting solution $\Phi_\mathrm{rep,0}$ is the repulsive polaron without Bogoliubov approximation. Since for small positive impurity-boson scattering lengths $a$ such that $a/\xi \ll 1$, the condensate distortion of the repulsive polaron relative to the unperturbed condensate is $\mathcal{O}(a/\xi)$ [61, 62], $\mathcal{H}_\mathrm{MF}[\Phi_\mathrm{rep,0}, \mathbb{I}]$ equals $\mathcal{H}_\mathrm{MF}[\sqrt{n_0}, \mathbb{I}]$

up to perturbative terms coming from the condensate distortion. Thus, the Bogoliubov transformation $S_\mathrm{rep,1}$ that diagonalizes $\mathcal{H}_\mathrm{MF}[\Phi_\mathrm{rep,0}, \mathbb{I}]$ is identical to the standard Bogoliubov transformation $S_\mathrm{Bog}$ of an unperturbed BEC, up to corrections of $\mathcal{O}((a/\xi)^2)$.

The next step correction to the repulsive polaron amounts to finding $\Phi_\mathrm{rep,1}$ such that $\zeta[\Phi_\mathrm{rep,1}, \Gamma_\mathrm{Bog}] = 0$. The differential equation $\zeta[\Phi, \Gamma_\mathrm{Bog}] = 0$ differs from $\zeta[\Phi, \mathbb{I}] = 0$ only in the terms containing $\Gamma_\mathrm{Bog}^{11}$ and $\Gamma_\mathrm{Bog}^{12}$, both of the order $\mathcal{O}(\lambda^{3/2}) \sim 5 \times 10^{-3}$, with $\lambda = n_0^{1/3} a_\mathrm{B}$ the BEC gas parameter [75, 76]. Due to the diluteness of cold atomic gases, $\lambda \ll 1$, and including bosonic correlations through $\Gamma$ within Bogoliubov approximation and beyond does not affect the repulsive polaron solution and the quantum fluctuations atop. Thus, in connection to the special setting we consider here, hereafter we neglect corrections due to quantum fluctuations of the repulsive polaron and set $S_\mathrm{rep} = \mathbb{I}$. Note that in general settings, especially pertaining to atomic BECs in lower dimensionality or exciton-polariton condensates in semiconductor heterostructures, it is essential to include the effects of quantum fluctuations through $\Gamma$, and our theory is capable to account for such effects in principle.

The next inputs to our variational theory are the bound state Bogoliubov factors $u_{\mathrm{B},\mathbf{x}}$ and $v_{\mathrm{B},\mathbf{x}}$, which form the bound state solution of $\mathcal{H}_\mathrm{MF}[\Phi, 0]$. It can be shown that the contribution of the off-diagonal terms in $\mathcal{H}_\mathrm{MF}[\Phi, 0]$ to the eigenstates and eigenenergies are of $\mathcal{O}(\mu/\varepsilon_\mathrm{B}) \sim 9 \times 10^{-3}$, and can be neglected to the leading order. This approximation amounts to setting $v_{\mathrm{B},\mathbf{x}} = 0$. Furthermore, the effective potential $U_0|\varphi_\mathrm{rep,\mathbf{x}}|^2 - \mu$ caused by the repulsive polaron's condensate distortion around the impurity is much weaker than $V_\mathrm{IB}(\mathbf{x})$, thus $u_{\mathrm{B},\mathbf{x}}$ can be approximated by $\eta_\mathbf{x}$ that is the bound state solution of $-\hbar^2\nabla^2/2m_\mathrm{red} + V_\mathrm{IB}(\mathbf{x})$ - see Appendix B for a detailed derivation of these perturbative approximations. Note that the leading-order approximations made above can be extended to arbitrary higher orders in a systematic manner, and we expect that the quantitative changes will not alter any of the key physics of the many-body bound states.

By carrying out the previous steps, we are in a position to obtain the metastable states from finding the optimal solutions of Eqs. 15, 16 and 17 by solving the variational equations (see Appendix C for the explicit form)

$$
\begin{aligned}
&\frac{\delta}{\delta\alpha_\mathbf{x}^*} H[\psi_n^*, \psi_n, \alpha_\mathbf{x}^*, \alpha_\mathbf{x}] - \lambda\eta_\mathbf{x} = 0\,, \\
&\frac{\delta}{\delta\psi_n^*} H[\psi_n^*, \psi_n, \alpha_\mathbf{x}^*, \alpha_\mathbf{x}] = E\,\psi_n\,.
\end{aligned}
\tag{19}
$$

In Eq. 19, $\lambda$ is a Lagrange multiplier determined to fulfill Eq. 17, and $E$ is the energy of the metastable state that also acts as a Lagrange multiplier to fulfill the normalization condition Eq. 16. Solving Eqs. 19 gives access to the energies and variational states of the many-body bound states across the Feshbach resonance, which are discussed in the next sections.

## A. Energy of the many-body bound states

In the regime $\mu/\varepsilon_B \ll 1$, we already noted that the condensate distortion $\alpha_{\mathbf{x}}$ remains small in magnitude compared to the repulsive polaron field $\varphi_{\rm rep}$, and as we will discuss at the end of this subsection, the energies and wave functions of the many-body bound states obtained by solving Eqs. 19 are well approximated by setting $\alpha_{\mathbf{x}} = 0$, meaning a vacuum of scattering Bogoliubov modes on top of the repulsive polaron. Fig. 2(a) depicts the energies of the metastable states obtained by setting $\alpha_{\mathbf{x}} = 0$ (note the unusual rescaling of the energy scale. Plots on linear scale are provided in Appendix D). In the attractive side (region (1) in Fig. 2(a)), the only stable-state solution corresponds to the attractive polaron $\Phi_{\rm att}$ (green dashed line), studied in Refs. [60, 62, 63]. All the fluctuation modes that are eigenstates of $\mathcal{H}_{\rm MF}[\Phi_{\rm att}, \mathbb{I}]$ have positive energy with a parabolic energy landscape as in panel (1) in Fig. 2(b).

On the repulsive side, there exists a range of scattering lengths where impurity-boson interactions lead to the instability of the phase quadrature of a Bogoliubov mode, leading to dynamical instability. The dynamical instability is a precursor to the formation of repulsive polaron, and occurs for a range of scattering lengths which lies inside the region (2) in Fig. 2(a). The energy landscape of the dynamically unstable mode is depicted in panel (2) of Fig. 2(b), where the negative- and positive-curvature directions correspond to the phase and amplitude quadratures, respectively.

In region (3) of Fig. 2(a), a well-defined unstable fluctuation mode emerges, that is the bound Bogoliubov mode. The possibility of multiple occupation of the bound Bogoliubov mode results in the emergence of the two metstable states $ms_1$ and $ms_2$, depicted by solid red and blue lines, respectively, in Fig. 2(a). The corresponding energy landscape in the form of a mexican hat, alongside the relative energies of various metastable states are depicted in panel (3) of Fig. 2(b). The origin of the energy landscape corresponds to the vaccum of the fluctuation mode, i.e. the repulsive polaron. The metastable states $ms_1$ and $ms_2$ are designated on the energy landscape schematically by circles whose radii and relative positions indicate the mean bound state occupation number and the relative energy of the states, respectively.

The energy landscape minimum corresponds roughly to the bound state component of the attractive polaron coherent state field, obtained by calculating the overlap $\alpha_{\rm att,B} = \int_{\mathbf{x}} \eta_{\mathbf{x}}^* \varphi_{\rm att,\mathbf{x}}$. In fact, we interpret the lowest-lying many-body bound state as *nothing but* the remnant of the attractive polaron branch on the repulsive side of the Feshbach resonance. The two variational states we employ here, i.e. the attrative polaron and the $ms_1$ state have similar but not identical structures, which explains their slightly different variational energies. To further support our claim, in Fig. 3 we compare density-profile of bosons around the impurity for the different variational

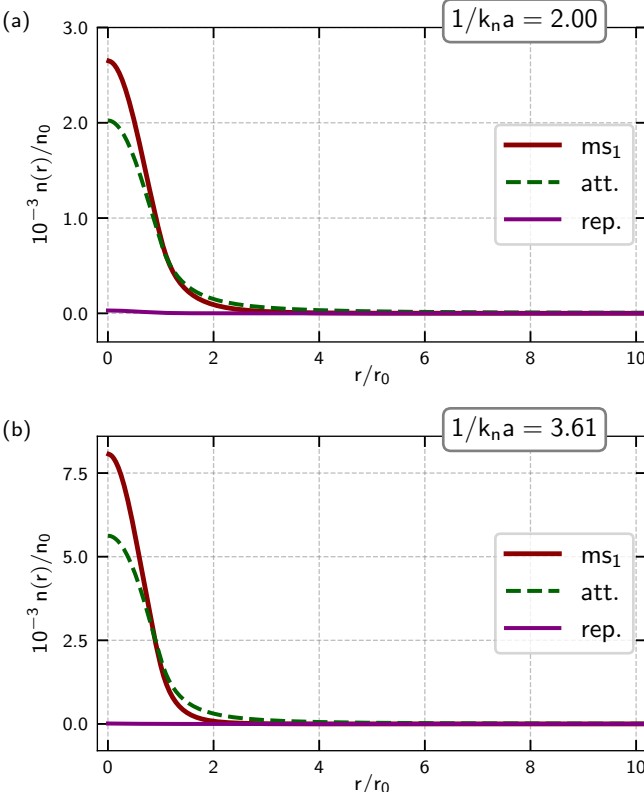

FIG. 3. Density profile of the repulsive polaron (solid purple line), attractive polaron (green dashed line), and $ms_1$ state (solid red line), as a function of the radial distance from the impurity, for (a) $1/k_n a = 2.0$ and (b) $1/k_n a = 3.61$. The density profiles of the attractive polaron and the $ms_1$ state are qualitatively similar.

states. The qualitative similarity of the spatial structures of the lowest-lying many-body bound state and the attractive polaron further suggests that the two states describe the same ground state.

Beyond a certain critical scattering length, new stable solutions emerge from the repulsive polaron, denoted by $ms_3$ to $ms_6$ in Fig. 2. These states correspond to multiple occupation of the bound state. As the interaction strength rises, the bound state becomes more localized, resulting in an increase in the effective inter-boson repulsive interaction. At the same time, the system gains energy by binding more bosons. While both these effects compete, the increase in bound state energy dominates, lowering the energy of the states with higher bound state occupation. In terms of the saddle point structure, the increase in bound state energy means that the saddle point gets deeper, and the mean occupation number of the bound state increases, as depicted in panel (4) of Fig 2(b). Another implication of the competition between the increase in binding energy and the repulsive interaction is the emergence of level crossings among the metastable states in region (4) of Fig. 2(a). The presence of such level crossings can be explained again by the simple model laid

out in the introduction. For a fixed bound state energy $\varepsilon_{\mathrm{B},0}$, two metastable states with $n_1$ and $n_2$ occupation of the bound state with $n^*(\varepsilon_{\mathrm{B},0}) < n_1 < n_2 < 2n^*(\varepsilon_{\mathrm{B},0})$ have energies $E_{n_1} < E_{n_2}$. For larger $1/k_n a$, the increase in binding energy has the dominant effect on the energy of the many-body bound states, and the energy of the state with higher bound state occupation decreases more rapidly, resulting in the level crossing pattern.

Fig. 4 depicts the behavior of energy and bound state occupation for the first few many-body bound states together with the attractive and repulsive polaron. The energy of the $ms_1$ state decreases monotonically, and its mean bound state occupation number saturates to double occupation for the range of scattering lengths considered. The $ms_2$ state approaches the bare dimer in energy and bound state occupation number. Across the level crossings of the two lowest-lying states, $\langle N_{\mathrm{B}} \rangle$ shows a non-monotonic behavior, and by increasing $1/k_n a$ saturates to single and double occupation for $ms_2$ and $ms_1$ states, respectively. The $ms_3$ state appears in the atom-dimer continuum at a critical scattering length (marked by the vertical dotted line in Fig. 4 (b)) and maintains a constant $\langle N_{\mathrm{B}} \rangle \simeq 3$. In contrast, the mean bound state occupation number of the attractive polaron increases monotonically with a value that remains larger than $ms_1$ and $ms_2$. At the level crossing of $ms_1$ and $ms_3$, the two states demonstrate strong mixing, resulting in spikes of $\langle N_{\mathrm{B}} \rangle$ for both states.

Before moving on to the next section, we comment on the approximation $\alpha_{\mathbf{x}} = 0$ introduced earlier. In Fig. 5, we compare the energies of many-body bound states obtained from solving the full set of Eqs. 19, to the energies obtained under the assumption $\alpha_{\mathbf{x}} = 0$. We find that the effect of condensate distortion on the wave functions and energies of many-body bound states are only marginal, and setting $\alpha_{\mathbf{x}} = 0$ is a reasonable approximation.

The main reason behind the markedly different behavior of the many-body bound states compared to the attractive and repulsive polaron lies in the particular composition of each many-body state $\left| \psi_{(\mathrm{B})} \right\rangle$ out of dimer Fock states $\{ |n\rangle_{\mathrm{B}}, n = 0, 1, 2, \cdots \}$. Indeed, inspection of $\langle N_{\mathrm{B}} \rangle$ in Fig. 4 suggests that $\left| \psi_{(\mathrm{B})} \right\rangle$ for each of the many-body bound states has to be close to a Fock state $|n\rangle_{\mathrm{B}}$ for some $n$. To gain further insight into the structure of the many-body bound states, in the next subsection, we investigate the dimer occupation statistics of the many-body bound states.

## B. Dimer occupation statistics of the many-body bound states

As mentioned at the end of Sec. II B 1, pure mean-field approaches to model the state of Bose polaron neglect the higher order terms $\hat{H}_3$ and $\hat{H}_4$, while the latter are crucial to capture the physics of many-body bound states. One consequence of including these higher order terms in the model is their non-perturbative effects reflected in

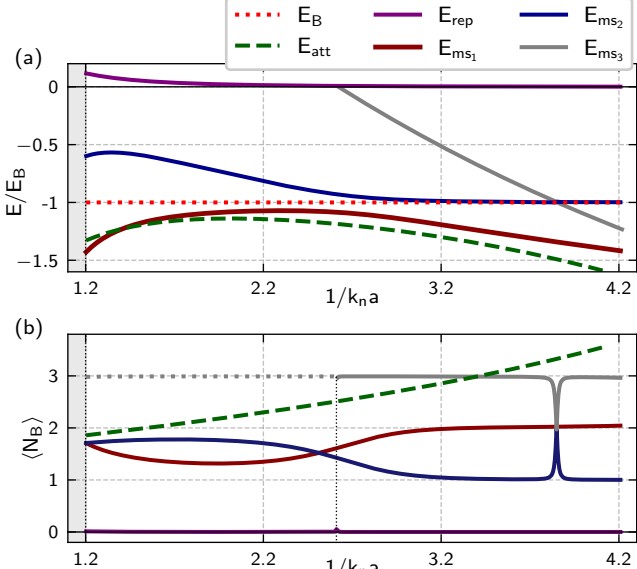

FIG. 4. Energy in units of the dimer binding energy (a) and mean bound state occupation number (b) of the many-body bound states (red, blue and grey solid lines for $ms_1$, $ms_2$ and $ms_3$ respectively), attractive polaron (green dashed line), and repulsive polaron (purple solid line). Initially, the $ms_2$ state has higher mean bound state occupation number and energy than the $ms_1$ state, indicating the dominant effect of the inter-boson interaction on the energy of the states. Beyond the first level crossing, the mean occupation number of the $ms_1$ state increases above the $ms_2$ state due to the gain in energy from binding more bosons. The $ms_3$ state enters the dimer-boson continuum at the critical scattering length indicated by vertical dotted line in panel (b) and maintains an almost constant $N_{\mathrm{B}} \simeq 3$. For increasing $1/k_n a$, the mean bound state occupation number of $ms_1$ and $ms_2$ states approach integer values. At the level crossing between $ms_1$ and $ms_3$, the states strongly mix, resulting in sharp spikes in $\langle N_{\mathrm{B}} \rangle$ in panel (b).

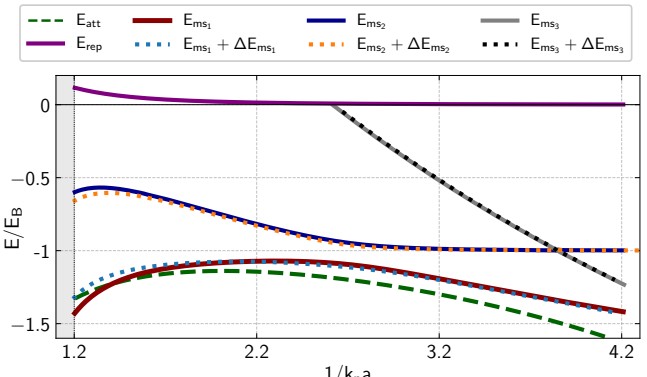

FIG. 5. Energy of the many-body bound states including the effect of condensate distortion obtained by fully solving Eqs. 19 (dotted lines), compared to the energies obtained by setting $\alpha_{\mathbf{x}} = 0$. Including condensate distortion effects results in marginal changes in the energy (denoted by $\Delta E_{ms_i}, i = 1, 2, 3$), and wave function of many-body bound states.

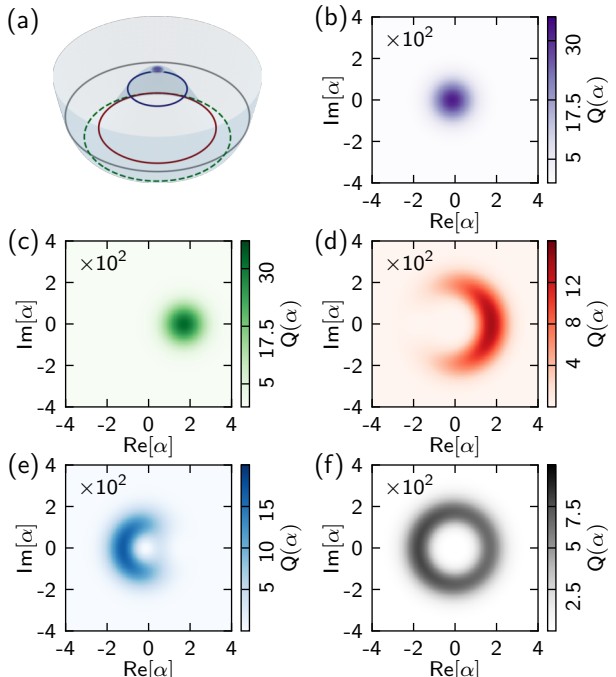

FIG. 6. (a) Illustration of the energy landscape and the metastable states at $1/k_n a = 2.74$. As in Fig. 2 (b) panels (3) and (4), the radius, respectively, the vertical order of each circle on the energy surface reflect the mean bound state occupation number, respectively, the energy of the corresponding metastable state. Panels (b) and (c) show the quantitative calculations of the Q representation of the repulsive and attractive polaron, respectively. Panels (d) to (f) depict the Q representation of $ms_1$ to $ms_3$ states.

the genuine quantum mechanical correlations of the wave function in the dimer Fock space, which is represented in our variational scheme by $\left|\psi_{(B)}\right\rangle$. To quantify the quantum mechanical correlations of $\left|\psi_{(B)}\right\rangle$, we note that it formally belongs to the Fock state of a single bosonic mode $\hat{b}$, thus its characteristics can be quantified via different quantum mechanical quasiprobability distributions used frequently in quantum optics to characterize the quantum states of light.

A quasiprobability distribution that is specially suitable for characterizing $\left|\psi_{(B)}\right\rangle$ is the Husimi Q representation, that in our context can be defined by [77]

$$Q(\alpha) = \frac{1}{\pi} \left\langle \alpha | \psi_{(B)} \right\rangle \left\langle \psi_{(B)} | \alpha \right\rangle . \tag{20}$$

In Eq. 20, $|\alpha\rangle$ is an arbitrary coherent state that is the eigenstate of $\hat{b}$, i.e. $\hat{b}|\alpha\rangle = \alpha|\alpha\rangle$.

In Fig. 6, we depict the Q representation of the states in Fig. 4 for $1/k_n a = 2.74$. The repulsive and attractive polaron, both include coherent state components of the bosonic mode $\hat{b}$ with a coherent state amplitude $\alpha^{(sp)} = \int_{\mathbf{x}} \eta_{\mathbf{x}}^* \varphi_{\mathbf{x}}^{(sp)}$ with the superscript "sp" indicating the respective saddle point. The Q representation of the saddle

point state is thus $Q^{(sp)}(\alpha) = 1/\pi \exp\left(-|\alpha - \alpha^{(sp)}|^2\right)$, which is a Gaussian distribution localized on $\alpha^{sp}$. In contrast, the many-body bound states have markedly different Q representations, reminiscent of Fock states. The Q representation already indicates that the state $\left|\psi_{(B)}\right\rangle$ contains quantum mechanical correlations with non-Gaussian characters, as opposed to coherent and squeezed coherent states that are characterized by ellipsoidal Q distributions. We again highlight that the non-Gaussianity of the Q distribution is a result of including higher order terms $\hat{H}_3$ and $\hat{H}_4$ in the model, and treating the boson correlations in the dimer Fock state sector exactly. Note that with the strong boson-boson repulsions considered here, a truncated-basis variational ansatz can be accurate enough to predict essential features of the polaron, however, it is best suited for the limit of low densities. Our theory, on the other hand, has the capability to include a fluctuating number of particles in the polaron cloud even in dense bosonic media, as long as the binding energy is much larger than the BEC chemical potential.

Another useful quantity signifying the correlations of bosons occupying the bound state is $g_B^{(2)}$ defined by

$$g_B^{(2)} = \frac{\left\langle \psi_{(B)} \right| \hat{b}^\dagger \hat{b}^\dagger \hat{b} \hat{b} \left| \psi_{(B)} \right\rangle}{\left\langle \psi_{(B)} \right| \hat{b}^\dagger \hat{b} \left| \psi_{(B)} \right\rangle^2} . \tag{21}$$

Fig. 7 depicts $g_B^{(2)}$ for different many-body bound states. We again observe that due to the effect of boson-boson repulsion, $g_B^{(2)}$ shows strong boson anti-bunching for all the many-body bound states. Especially, the states beyond $ms_2$ have $g_B^{(2)} \simeq 1 - 1/n$ with $n \geq 3$, a hallmark signature of Fock states in contrast to coherent states that have $g^{(2)}(0) = 1$.

## C. Spectral signatures of the many-body bound states

Here we consider the experimental observability of the many-body bound states we predicted above. An experimentally relevant quantity in polaron spectroscopy is the quasiparticle residue, defined as

$$Z(E) = \sum_i |\langle GS_0 | i \rangle|^2 \delta(E - E_i), \tag{22}$$

where $|i\rangle$ is an eigenstate of the interacting system with energy $E_i$, and $|GS_0\rangle$ is the non-interacting ground state. In the case of Bose polarons, the non-interacting ground state consists of an impurity and an unperturbed condensate with no mutual interactions. In contrast, the interacting state is of the form $\hat{\mathcal{O}}_i |GS\rangle$, where $\hat{\mathcal{O}}_i$ creates the appropriate excitations of the eigenstate $i$ on top of the interacting ground state.

In Fig. 8 (a), the variation of $Z$ across the Feshbach resonance is depicted for each stable states, as well as

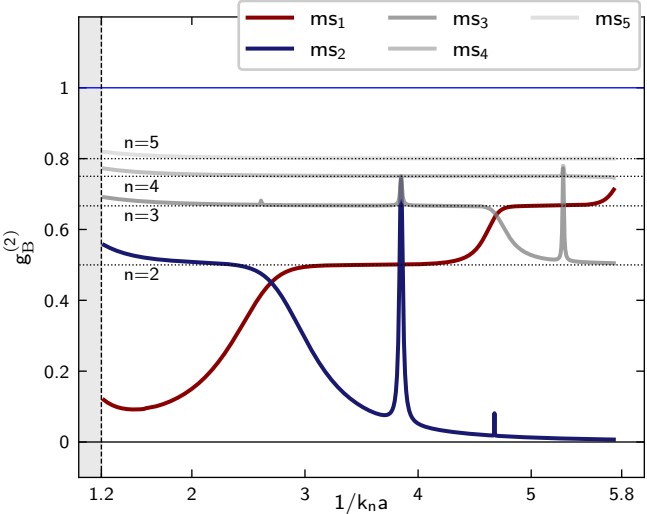

FIG. 7. $g_{\mathrm{B}}^{(2)}$ of the many-body bound states. Clear deviations from the results of a Gaussian state indicates the non-Gaussian nature of bosons spatial correlations occupying the bound state.

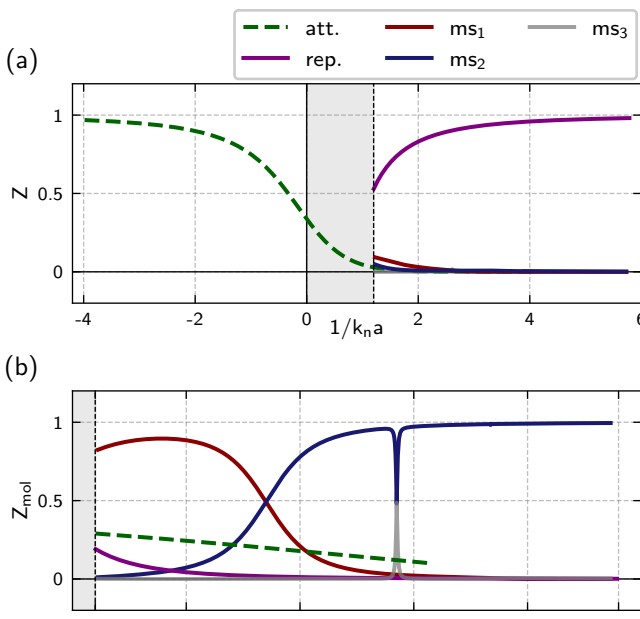

FIG. 8. (a) Quasiparticle residue of different many-body bound states, compared to the attractive and repulsive polaron. At strong couplings, the quasiparticle residue of attractive polaron and all the many-body bound states are substantially smaller than the repulsive polaron for strong couplings. (b) Molecular quasiparticle residue of the states in (a). The states $ms_1$ and $ms_2$ have substantial molecular weight with non-monotonic behavior as a function of $1/k_n a$, in contrast to the prediction for the attractive polaron. The sharp spikes in $Z_{\mathrm{mol}}$ of $ms_1$ and $ms_3$ occurs at the corresponding level crossing.

the $Z$ factor for attractive and repulsive polaron. We observe that although the quasiparticle weights of $ms_1$ and $ms_2$ states are higher than the attractive polaron, all the other many-body states have essentially vanishing quasiparticle residue. This observation is compatible with the conclusion that beyond $ms_2$, the many-body states are well characterized by Fock states $|n\rangle_{\mathrm{B}}$ for $n \geq 3$ with vanishing quasiparticle residue. Furthermore, as the repulsive inter-boson interaction is decreased, the $Z$ factor of attractive polaron and all the many-body bound state excitations decrease due to an increasing number of bound state excitations.

Furthermore, in connection with detecting molecular spectra in ultracold mixtures, a molecular quasiparticle residue can be defined as

$$Z_{\mathrm{mol}}(E) = \sum_n |\langle \mathrm{GS_{mol}}|n\rangle|^2 \, \delta(E - E_n), \qquad (23)$$

where $|\mathrm{GS_{mol}}\rangle$ is a state comprised of an unperturbed condensate and a single impurity-boson dimer. This quasiparticle residue is suggested in [78, 79] to detect molarons and observe polaron-molecule transition in impurity-Fermi systems. Fig. 8(b) shows $Z_{\mathrm{mol}}$ of the many-body bound states. Interestingly, $ms_1$ and $ms_2$ states have substantial $Z_{\mathrm{mol}}$, with the non-monotonous variation with $1/k_n a$ compatible with their bound state occupation number. For the attractive polaron, the magnitude of $Z_{\mathrm{mol}}$ is of the same order of magnitude as $ms_1$, although quantitative differences point to the remaining differences of these variational states. Thus, $Z_{\mathrm{mol}}$ can be a sensitive probe for detection of many-body bound states and to elucidate the exact nature of the overall ground state.

The $ms_3$ state exhibits a vanishing $Z_{\mathrm{mol}}$ except for values of $1/k_n a$ close to the level crossing with $ms_1$ state, where $Z_{\mathrm{mol}}$ of both states vary rapidly and coincide at $Z_{\mathrm{mol}} = 0.5$.

## IV. COMPARISON TO THE EXISTING METHODS

As mentioned earlier, the crucial assumption of the variational formalism developed in this work is the large separation of energy scales between the dimer binding energy $\varepsilon_{\mathrm{B}}$ and the typical energy of the Bogoliubov excitations (of the order of $\mu$). This condition is violated close to the unitarity on the repulsive side. The other important assumption concerns the existence of a well-defined unstable Bogoliubov mode on top of the repulsive polaron saddle point, which breaks down in the presence of a dynamical instability.

Variational schemes such as truncated basis methods or Gaussian state theories including boson-boson inteactions are in principle able to surpass these limitations. Truncated basis methods are able to give access to the full excitation spectrum and include multi-body correlations exactly, however, they are limited in the number

of particles included in the variational state. In comparison, our approach includes exact correlations only among excitations bound to the impurity and neglects some correlations of excited scattering states, that is suitable for heavy impurities. Nevertheless, it does not restrict the number of excitations included in the ansatz. Gaussian state theories are able to access the exact stable saddle point of the system by optimizing $\Phi$ and $\Gamma$. However, the states with non-Gaussian correlations are not included in the variational manifold.

An improvement to our ansatz is to include Bogoliubov transformation as a variational parameter, and obtain the modifications of the Bogoliubov spectrum due to the presence of the impurity. This approach has already been incorporated to study the modification of local boson correlations in the vicinity of the impurity [52, 53], and predicted many-body shifts of Efimov states. Including these correlations in our ansatz partially accounts for three-body correlations on a many-body level. However, it is a numerically challenging task to obtain metastable variational solutions and we leave this problem for future research.

## V. CONCLUSION AND OUTLOOK

In this work, we addressed the problem of Bose polaron at strong couplings. We introduced a variational scheme that is suitable for the regime when the impurity-boson binding energy is much larger than the BEC chemical potential. We presented a comprehensive theoretical formalism that is sufficiently general to be applicable to dilute and dense bosonic media in any dimensions, ranging from ultracold atomic mixtures to excitonic condensates in semiconductor heterostructures, and include effects that are crucial to describe Bose polarons at strong couplings. We demonstrated that the interplay of impurity-induced instability and repulsive inter-boson interactions leads to the existence of multiple metastable states in the form of many-body bound states with intermediate energies lying between the attractive and repulsive polaron.

Crucially, the existence and properties of the many-body bound states we predict are closely linked to the non-perturbative nature of the problem captured by the higher order interaction processes $\hat{H}_3$ and $\hat{H}_4$, involving three- and four-boson terms, respectively. Within our variational approach, we showed that including the resulting correlations among the bound bosons exactly leads to the emergence of genuine quantum mechanical characteristics of the wave function, especially non-Gaussian correlations and interaction-induced antibunching. Furthermore, these many-body bound states can have observable signatures in molecular spectroscopy techniques with quasiparticle weights considerably different from the coherent state theory prediction for the attractive and repulsive polarons.

The theoretical developments in this work present one natural scheme to separate the modes of the strong coupling impurity-boson system into a few strongly interacting modes requiring non-perturbative treatment, and a continuum of weakly interacting modes. With this theory we are able to explore a broad range of parameters and map out the phase diagram of the strong coupling Bose polaron. In particular, we clarified how the attractive polaron continuously evolves into a multi-body bound state as one crosses the Feshbach resonance into the repulsive side. Thereby we arrive at a unified theory of repulsive and attractive Bose polarons.

One future direction concerns studying the influence of few-body Efimov-type correlations on the properties of many-body bound states. In particular, establishing universal features of many-body bound states and clarifying the role of finite range effects constitute important open problems.

In the present context, we pointed out the crucial role of phonon nonlinearities on the physics of strong coupling Bose polarons. It would be interesting to expand the scope of this work by considering other models where phonon nonlinearities play a crucial role, for instance, to study impurity motion in nonlinear bosonic models with non-perturbative solitonic excitations (e.g. in the Ferenkel-Kontorova model [80], or models described by the nonlinear Schrodinger equation [81]). As another avenue, one could apply this framework to study the motion of single holes in quantum antiferromagnets [82–84] or the formation of magnon-impurity bound states [85].

## ACKNOWLEDGMENTS

We thank Richard Schmidt, Arthur Christianen, Clemens Kuhlenkamp, Ataç İmamoğlu, Monika Aidelsburger, Jean Dalibard, Luis A. Peña Ardila, Kristian K. Nielsen, Meera M. Parish, Jesper Levinsen, Eugene Demler, Christoph Eigen, Zoran Hadzibabic, Timour Ichmoukhamedov, Tobias Graß, Alek Bedroya, Felix A. Palm, Marcel Gievers, Nepomuk Ritz and Oriana K. Dießel for insightful discussions. We thank Pietro Massignan for pointing out the connection between the solutions of the GP equation and the polaronic branches. N. M. and F. G. acknowledge funding by the Deutsche Forschungsgemeinschaft (DFG, German Research Foundation) under Germany's Excellence Strategy – EXC-2111 – 390814868 and from the European Research Council (ERC) under the European Union's Horizon 2020 research and innovation programm (Grant Agreement no 948141) — ERC Starting Grant SimUcQuam. Work in Brussels is supported by the FRS-FNRS (Belgium), the ERC Starting Grants TopoCold and LATIS, and the EOS project CHEQS.

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

## Appendix A: Mean-field decoupling of $\hat{H}_{\mathrm{LLP}}$

Here we detail on the mean-field decoupling procedure. Using the Wick's theorem [86], for $\hat{H}_{\mathrm{LLP}}$ in Eq. 4, the mean-field Hamiltonian takes the following form,

$$
\begin{aligned}
\hat{H}_{\mathrm{LLP}} = E &+ \left( \delta\hat{\Psi}^{\dagger} \cdot \zeta + h.c. \right) \\
&+ \frac{1}{2} : \delta\hat{\Psi}^{\dagger} \mathcal{H}_{\mathrm{MF}} \delta\hat{\Psi} : + \hat{H}_3 + \hat{H}_4 \,.
\end{aligned}
\tag{A1}
$$

with explicit expressions for different terms as the following,

$$
\begin{aligned}
E = \frac{\hbar^2 \mathbf{K}_0^2}{2M} &- \frac{\hbar \mathbf{K}_0}{M} \cdot \left( \int_{\mathbf{x}} \varphi_{\mathbf{x}}^* (-i\hbar\nabla_{\mathbf{x}})\varphi_{\mathbf{x}} + \int_{\mathbf{k}} \hbar\mathbf{k}\Gamma_{\mathbf{kk}}^{11} \right) + \int \left( \frac{\hbar\mathbf{k} \cdot \hbar\mathbf{k}'}{2M} \right) \left\{ \Gamma_{\mathbf{kk}}^{11}\Gamma_{\mathbf{k'k'}}^{11} + |\Gamma_{\mathbf{kk'}}^{11}|^2 + |\Gamma_{\mathbf{kk'}}^{12}|^2 \right. \\
&+ \left( \varphi_{\mathbf{k}}\varphi_{\mathbf{k'}}\Gamma_{\mathbf{k'k}}^{21} + \varphi_{\mathbf{k'}}^*\varphi_{\mathbf{k}}\Gamma_{\mathbf{kk'}}^{11} + \varphi_{\mathbf{k'}}^*\varphi_{\mathbf{k'}}\Gamma_{\mathbf{kk}}^{11} + c.c. \right) + |\varphi_{\mathbf{k}}|^2 |\varphi_{\mathbf{k'}}|^2 \Big\} + \int_{\mathbf{x}} \varphi_{\mathbf{x}}^* \left( -\frac{\hbar^2\nabla^2}{2m_{\mathrm{red}}} + V_{\mathrm{IB}}(\mathbf{x}) - \mu \right) \varphi_{\mathbf{x}} \\
&+ \int_{\mathbf{x}} \left( -\frac{\hbar^2\nabla^2}{2m_{\mathrm{red}}} + V_{\mathrm{IB}}(\mathbf{x}) - \mu \right) \Gamma_{\mathbf{xx}}^{11} + \frac{1}{2}\int_{\mathbf{x},\mathbf{x}'} U_{\mathrm{BB}}(\mathbf{x}-\mathbf{x}') \Big\{ |\varphi_{\mathbf{x}}|^2 |\varphi_{\mathbf{x'}}|^2 + (\varphi_{\mathbf{x}}^*\varphi_{\mathbf{x'}}^*\Gamma_{\mathbf{xx'}}^{12} \\
&+ |\varphi_{\mathbf{x}}|^2\Gamma_{\mathbf{x'x'}}^{11} + \varphi_{\mathbf{x'}}^*\varphi_{\mathbf{x}}\Gamma_{\mathbf{xx'}}^{11} + c.c.) + |\Gamma_{\mathbf{xx'}}^{12}|^2 + |\Gamma_{\mathbf{xx'}}^{11}|^2 + \Gamma_{\mathbf{xx}}^{11}\Gamma_{\mathbf{x'x'}}^{11} \Big\} \,.
\end{aligned}
\tag{A2}
$$

The linear Hamiltonian $\hat{H}_1$ has the following form,

$$
\hat{H}_1 = \int_{\mathbf{x}} \hat{\phi}_{\mathbf{x}}^{\dagger} \zeta_{\mathbf{x}} + h.c. \,,
\tag{A3}
$$

where we explicitly write the coordinate space integration instead of shorthand inner product. The vector $\zeta_{\mathbf{x}}$ then reads as

$$
\begin{aligned}
\zeta_{\mathbf{x}} = h_0\varphi_{\mathbf{x}} &+ \left[ \int_{\mathbf{x}'} U_{\mathrm{BB}}(\mathbf{x}-\mathbf{x}')\left( |\varphi_{\mathbf{x'}}|^2 + \Gamma_{\mathbf{x'x'}}^{11} \right) \right]\varphi_{\mathbf{x}} + \int_{\mathbf{x}'} \left\{ \left[ U_{\mathrm{BB}}(\mathbf{x}-\mathbf{x}') - \frac{1}{M}(-i\hbar\nabla_{\mathbf{x}}) \cdot (-i\hbar\nabla_{\mathbf{x'}}) \right]\Gamma_{\mathbf{x'x}}^{11} \right\}\varphi_{\mathbf{x'}} \\
&+ \int_{\mathbf{x}'} \left\{ \left[ U_{\mathrm{BB}}(\mathbf{x}-\mathbf{x}') + \frac{1}{M}(-i\hbar\nabla_{\mathbf{x}}) \cdot (-i\hbar\nabla_{\mathbf{x'}}) \right]\Gamma_{\mathbf{xx'}}^{12} \right\}\varphi_{\mathbf{x'}}^* + \int_{\mathbf{x}'} \frac{1}{M}\left[ -i\hbar\nabla_{\mathbf{x'}}\left( |\varphi_{\mathbf{x'}}|^2 + \Gamma_{\mathbf{x'x'}}^{11} \right) \right] \cdot (-i\hbar\nabla_{\mathbf{x}}\varphi_{\mathbf{x}}) \,,
\end{aligned}
\tag{A4}
$$

where $h_0 = -\hbar^2\nabla^2/2m_{\mathrm{red}} + V_{\mathrm{IB}}(\mathbf{x}) - \mu$. With $\zeta_{\mathbf{x}}$ as in Eq. A4, the saddle point condition is $\zeta_{\mathbf{x}} = 0$. In

the special case of $\Gamma_{\mathbf{xx'}}^{11} = \Gamma_{\mathbf{xx'}}^{12} = 0$ and $M \to \infty$, the saddle point condition reduces to the Gross-Pitaevskii

equation for the condensate, including the distortion caused by the impurity (that is encoded in the impurity-boson potential in $h_0$). Given the saddle point condition $\zeta_\mathbf{x} = 0$, and the boundary condition on the condensate that $\lim_{|\mathbf{x}|\to\infty} \varphi_\mathbf{x} = \sqrt{n_0}$, the chemical potential including the Lee-Huang-Yang and finite boson-boson range corrections reads as

$$\mu = (n_0 + \Gamma_{00}^{11}) \int_\mathbf{x} U_{\rm BB}(\mathbf{x}) + \int_\mathbf{x} U_{\rm BB}(\mathbf{x}) \mathrm{Re}(\Gamma_{\mathbf{x}0}^{12} + \Gamma_{\mathbf{x}0}^{11}). \tag{A5}$$

The quadratic Hamiltonian is of the following form

$$\hat{H}_2 = \frac{1}{2} : \int_{\mathbf{k},\mathbf{k}'} \left( \delta\hat{\phi}_{\mathbf{k}'}^\dagger \quad \delta\hat{\phi}_{-\mathbf{k}'} \right) \mathcal{H}_{\mathbf{k}'\mathbf{k}}^{\rm imp} \begin{pmatrix} \delta\hat{\phi}_\mathbf{k} \\ \delta\hat{\phi}_{-\mathbf{k}}^\dagger \end{pmatrix} :$$
$$+ \frac{1}{2} : \int_{\mathbf{x},\mathbf{x}'} \left( \delta\hat{\phi}_{\mathbf{x}'}^\dagger, \delta\hat{\phi}_{\mathbf{x}'} \right) \mathcal{H}_{\mathbf{x}'\mathbf{x}} \begin{pmatrix} \delta\hat{\phi}_\mathbf{x} \\ \delta\hat{\phi}_\mathbf{x}^\dagger \end{pmatrix} :, \tag{A6}$$

thus, $\mathcal{H}_{\rm MF}$ consists of two terms: $\mathcal{H}^{\rm imp}$ comes from the finite mass of the impurity, and $\mathcal{H}$ is the mean-field Hamiltonian in the limit $M \to \infty$. The explicit forms of $\mathcal{H}$ and $\mathcal{H}^{\rm imp}$ are of the following form,

$$\mathcal{H}_{\mathbf{k}'\mathbf{k}}^{\rm imp} = \begin{pmatrix} \mathcal{E}_{\mathbf{k}'\mathbf{k}}^{\rm imp} & \Delta_{\mathbf{k}'\mathbf{k}}^{\rm imp} \\ \Delta_{(-\mathbf{k}')(-\mathbf{k})}^{\rm imp*} & \mathcal{E}_{(-\mathbf{k}')(-\mathbf{k})}^{\rm imp*} \end{pmatrix}, \tag{A7}$$

$$\mathcal{H}_{\mathbf{x}'\mathbf{x}} = \begin{pmatrix} \mathcal{E}_{\mathbf{x}'\mathbf{x}} & \Delta_{\mathbf{x}'\mathbf{x}} \\ \Delta_{\mathbf{x}'\mathbf{x}}^* & \mathcal{E}_{\mathbf{x}'\mathbf{x}}^*, \end{pmatrix}, \tag{A8}$$

where the diagonal and off-diagonal terms of $\mathcal{H}_{\rm imp}$ are as follows,

$$\mathcal{E}_{\mathbf{k}'\mathbf{k}}^{\rm imp} = \frac{\hbar\mathbf{k} \cdot \hbar\mathbf{k}'}{M} \left( \Gamma_{\mathbf{k}\mathbf{k}'}^{11} + \varphi_\mathbf{k}^* \varphi_{\mathbf{k}'} \right)$$
$$+ \delta^{(d)}(\mathbf{k} - \mathbf{k}') \int_{\mathbf{k}''} \frac{\hbar\mathbf{k}' \cdot \hbar\mathbf{k}''}{M} \left( \Gamma_{\mathbf{k}''\mathbf{k}''}^{11} + \varphi_{\mathbf{k}''}^* \varphi_{\mathbf{k}''} \right), \tag{A9}$$

$$\Delta_{\mathbf{k}'\mathbf{k}}^{\rm imp} = -\frac{\hbar\mathbf{k} \cdot \hbar\mathbf{k}'}{M} \left( \Gamma_{\mathbf{k}'(-\mathbf{k})}^{12} + \varphi_{\mathbf{k}'} \varphi_{-\mathbf{k}} \right). \tag{A10}$$

The diagonal and off-diagonal terms in $\mathcal{H}$ read as

$$\mathcal{E}_{\mathbf{x}'\mathbf{x}} = \delta^{(d)}(\mathbf{x} - \mathbf{x}') \Big[ h_0$$
$$+ \int_{\mathbf{x}''} U_{\rm BB}(\mathbf{x}' - \mathbf{x}'') \left( \Gamma_{\mathbf{x}''\mathbf{x}''}^{11} + |\varphi_{\mathbf{x}''}|^2 \right) \Big] \tag{A11}$$
$$+ U_{\rm BB}(\mathbf{x} - \mathbf{x}') \left( \Gamma_{\mathbf{x}\mathbf{x}'}^{11} + \varphi_\mathbf{x}^* \varphi_{\mathbf{x}'} \right),$$

$$\Delta_{\mathbf{x}'\mathbf{x}} = U_{\rm BB}(\mathbf{x} - \mathbf{x}') \left( \Gamma_{\mathbf{x}'\mathbf{x}}^{12} + \varphi_{\mathbf{x}'} \varphi_\mathbf{x} \right). \tag{A12}$$

Finally, the cubic and quartic terms are of the following forms

$$\hat{H}_3 = \int_{\mathbf{k},\mathbf{k}'} \frac{\hbar\mathbf{k} \cdot \hbar\mathbf{k}'}{M} \left( \varphi_\mathbf{k} : \delta\hat{\phi}_\mathbf{k}^\dagger \delta\hat{\phi}_{\mathbf{k}'}^\dagger \delta\hat{\phi}_{\mathbf{k}'} : + h.c. \right)$$
$$+ \int_{\mathbf{x},\mathbf{x}'} U_{\rm BB}(\mathbf{x} - \mathbf{x}') \left( \varphi_\mathbf{x} : \delta\hat{\phi}_\mathbf{x}^\dagger \delta\hat{\phi}_{\mathbf{x}'}^\dagger \delta\hat{\phi}_{\mathbf{x}'} : + h.c. \right). \tag{A13}$$

The quartic term representing the interaction of fluctuation modes reads as

$$\hat{H}_4 = \int_{\mathbf{k},\mathbf{k}'} \frac{\hbar\mathbf{k} \cdot \hbar\mathbf{k}'}{2M} : \delta\hat{\phi}_\mathbf{k}^\dagger \delta\hat{\phi}_{\mathbf{k}'}^\dagger \delta\hat{\phi}_{\mathbf{k}'} \delta\hat{\phi}_\mathbf{k} :$$
$$+ \frac{1}{2} \int_{\mathbf{x},\mathbf{x}'} U_{\rm BB}(\mathbf{x} - \mathbf{x}') : \delta\hat{\phi}_\mathbf{x}^\dagger \delta\hat{\phi}_{\mathbf{x}'}^\dagger \delta\hat{\phi}_{\mathbf{x}'} \delta\hat{\phi}_\mathbf{x} :. \tag{A14}$$

## Appendix B: Connection of the bound Bogoliubov mode to the bare impurity-boson bound state

Here we try to find the bound state of the quadratic Hamiltonian Eq. A6. As mentioned in the text, the exact excitation spectrum of the system is determined by finding $\varphi_{0,\mathbf{x}}$ and $S_{0,\mathbf{xy}}$ such that $\zeta[\Phi_0, \Gamma_0] = 0$ and $S_0^\dagger \mathcal{H}_{\rm MF}[\Phi_0, \Gamma_0] S_0 = \mathbb{I}_2 \otimes D$, while fulfilling $2\Gamma_0 + \mathbb{I} = S_0 S_0^\dagger$. The self-consistent solution can be obtained iteratively, starting from an unperturbed weakly-interacting Bose gas $\varphi_\mathbf{x}^{i=0} = \sqrt{n_0}$ and $S^{i=0} = \mathbb{I}$ as initial guess. At each step, the updated condensate field $\Phi_\mathbf{x}^{i+1} = (\varphi_\mathbf{x}^{i+1}, \varphi_\mathbf{x}^{i+1*})^{\rm T}$ satisfies $\zeta[\Phi^{i+1}, \Gamma^i] = 0$, and $S^{i+1}$ diagonalizes $\mathcal{H}_{\rm MF}[\Phi^i, \Gamma^i]$, giving the updated covariance matrix $\Gamma^{i+1}$. Iterations are then carried out until convergence.

In the first iteration, the quadratic Hamiltonian is

$$\hat{H}_2^{i=0} = \frac{1}{2} : \int_\mathbf{k} \left( \delta\hat{\phi}_\mathbf{k}^\dagger \quad \delta\hat{\phi}_{-\mathbf{k}} \right) \mathcal{H}_{\rm Bog}(\mathbf{k}) \begin{pmatrix} \delta\hat{\phi}_\mathbf{k} \\ \delta\hat{\phi}_{-\mathbf{k}}^\dagger \end{pmatrix} :$$
$$+ \frac{1}{2} : \int_{\mathbf{k},\mathbf{k}'} \left( \delta\hat{\phi}_{\mathbf{k}'}^\dagger \quad \delta\hat{\phi}_{-\mathbf{k}'} \right) \tilde{V}_{\rm IB}(\mathbf{k}' - \mathbf{k}) \mathbb{I}_{2\times2} \begin{pmatrix} \delta\hat{\phi}_\mathbf{k} \\ \delta\hat{\phi}_{-\mathbf{k}}^\dagger \end{pmatrix} :, \tag{B1}$$

where $\mathcal{H}_{\rm Bog}(\mathbf{k})$ is the standard Bogoliubov Hamiltonian

$$\mathcal{H}_{\rm Bog}(\mathbf{k}) = \begin{pmatrix} \epsilon_\mathbf{k} + n_0 U_{\rm BB}(\mathbf{k}) & n_0 U_{\rm BB}(\mathbf{k}) \\ n_0 U_{\rm BB}(\mathbf{k}) & \epsilon_\mathbf{k} + n_0 U_{\rm BB}(\mathbf{k}) \end{pmatrix}, \tag{B2}$$

with $\epsilon_\mathbf{k} = \hbar^2 \mathbf{k}^2 / 2m_{\rm red}$, and $\tilde{V}_{\rm IB}(\mathbf{k})$ is the Fourier transform of $V_{\rm IB}(\mathbf{x})$. $\mathcal{H}_{\rm Bog}(\mathbf{k})$ is diagonalized by the matrix $S_\mathbf{k}$ given by

$$S_\mathbf{k} = \begin{pmatrix} u_\mathbf{k} & -v_\mathbf{k} \\ -v_\mathbf{k} & u_\mathbf{k} \end{pmatrix} \tag{B3}$$

where $u_\mathbf{k} = \cosh(\theta_\mathbf{k})$, $v_\mathbf{k} = \sinh(\theta_\mathbf{k})$, and $\tanh(2\theta_\mathbf{k}) = n_0 U_{\rm BB}(\mathbf{k}) / (\epsilon_\mathbf{k} + n_0 U_{\rm BB}(\mathbf{k}))$. Diagonalization by $S_\mathbf{k}$ leads to the Bogoliubov dispersion relation

$$\varepsilon_\mathbf{k} = \sqrt{\epsilon_\mathbf{k} (\epsilon_\mathbf{k} + 2n_0 U_{\rm BB}(\mathbf{k}))}. \tag{B4}$$

The bound state of the Hamiltonian in Eq. B1 is obtained from

$$\begin{pmatrix} \epsilon_\mathbf{k} + n_0 U_{\rm BB}(\mathbf{k}) & n_0 U_{\rm BB}(\mathbf{k}) \\ n_0 U_{\rm BB}(\mathbf{k}) & \epsilon_\mathbf{k} + n_0 U_{\rm BB}(\mathbf{k}) \end{pmatrix} \begin{pmatrix} u_{\rm B,\mathbf{k}} \\ v_{\rm B,\mathbf{k}} \end{pmatrix}$$
$$+ \int_{\mathbf{k}'} \tilde{V}_{\rm IB}(\mathbf{k} - \mathbf{k}') \begin{pmatrix} u_{\rm B,\mathbf{k}'} \\ v_{\rm B,\mathbf{k}'} \end{pmatrix} = -\varepsilon_{\rm B} \begin{pmatrix} u_{\rm B,\mathbf{k}} \\ v_{\rm B,\mathbf{k}} \end{pmatrix}. \tag{B5}$$

Formally solving for $v_{B,\mathbf{k}}$ in Eq. B5 results in

$$v_{B,\mathbf{k}} = \int_{\mathbf{k}'} G(-\varepsilon_B)_{\mathbf{k}\mathbf{k}'} \, n_0 U_{BB}(\mathbf{k}') \, u_{B,\mathbf{k}'} \,, \qquad \text{(B6)}$$

where $G^{-1}(E)_{\mathbf{k}\mathbf{k}'} = \left(E - \epsilon_{\mathbf{k}} - n_0 U_{BB}(\mathbf{k})\right)\delta^{(d)}(\mathbf{k} - \mathbf{k}') - \tilde{V}_{IB}(\mathbf{k} - \mathbf{k}')$. Inserting $v_{B,\mathbf{k}}$ of Eq. B6 back in the equation satisfied by $u_{B,\mathbf{k}}$ results in

$$\left(\epsilon_{\mathbf{k}} + n_0 U_{BB}(\mathbf{k})\right) u_{B,\mathbf{k}} + \int_{\mathbf{k}'} \tilde{V}_{IB}(\mathbf{k} - \mathbf{k}') \, u_{B,\mathbf{k}'}$$
$$+ n_0 U_{BB}(\mathbf{k}) \int_{\mathbf{k}'} G(-\varepsilon_B)_{\mathbf{k}\mathbf{k}'} \, n_0 U_{BB}(\mathbf{k}') \, u_{B,\mathbf{k}'} = -\varepsilon_B \, u_{B,\mathbf{k}} \,.$$
$$\text{(B7)}$$

Applying standard perturbation theory to Eq. B7 in

the regime $n_0 U_{BB}(0) \ll \varepsilon_B$, $u_{B,\mathbf{k}}$ is obtained as the bound state of $-\hbar^2 \nabla^2 / 2 m_{red} + V_{IB}(\mathbf{x})$ up to corrections of $\mathcal{O}(n_0 U_{BB}(0)/\varepsilon_B)$. Thus, to leading order in $n_0 U_{BB}(0)/\varepsilon_B$, $u_{B,\mathbf{x}} = \eta_{\mathbf{x}}$ and $v_{B,\mathbf{x}} = 0$.

## Appendix C: Explicit form of variational equations

In this appendix, for the sake of completeness, we first derive the general form of variational equations in 19 for the case $\Gamma = 0$. Then we specialize the variational equations solved to obtain the variational states and energies of the many-body bound states presented in this work.

The coherent state $\alpha_{\mathbf{x}}$ satisfies the following nonlinear integro-differential equation

$$\begin{aligned}
&\left[h_0 + \int_{\mathbf{x}'} U_{BB}(\mathbf{x} - \mathbf{x}')|\varphi_{rep,\mathbf{x}'}|^2\right]\alpha_{\mathbf{x}} + \int_{\mathbf{x}'} U_{BB}(\mathbf{x} - \mathbf{x}') \, \varphi_{rep,\mathbf{x}}^* \varphi_{rep,\mathbf{x}'} \, \alpha_{\mathbf{x}'} + \int_{\mathbf{x}'} U_{BB}(\mathbf{x} - \mathbf{x}') \, \varphi_{rep,\mathbf{x}'} \varphi_{rep,\mathbf{x}} \, \alpha_{\mathbf{x}'}^* \\
&+ \int_{\mathbf{x}'} U_{BB}(\mathbf{x} - \mathbf{x}')\left[\tilde{\varphi}_{\mathbf{x}'}^* \tilde{\varphi}_{\mathbf{x}} - \varphi_{rep,\mathbf{x}'}^* \varphi_{rep,\mathbf{x}} + \Delta\langle: \hat{\phi}_{\mathbf{x}'}^{(B)\dagger} \hat{\phi}_{\mathbf{x}}^{(B)} :\rangle\right]\alpha_{\mathbf{x}'} \\
&+ \int_{\mathbf{x}'} U_{BB}(\mathbf{x} - \mathbf{x}')\left[|\tilde{\varphi}_{\mathbf{x}'}|^2 - |\varphi_{rep,\mathbf{x}'}^*|^2 + \Delta\langle: \hat{\phi}_{\mathbf{x}'}^{(B)\dagger} \hat{\phi}_{\mathbf{x}'}^{(B)} :\rangle\right]\alpha_{\mathbf{x}} \\
&+ \int_{\mathbf{x}'} U_{BB}(\mathbf{x} - \mathbf{x}')\left[\tilde{\varphi}_{\mathbf{x}'} \tilde{\varphi}_{\mathbf{x}} - \varphi_{rep,\mathbf{x}'} \varphi_{rep,\mathbf{x}} + \Delta\langle: \hat{\phi}_{\mathbf{x}'}^{(B)} \hat{\phi}_{\mathbf{x}}^{(B)} :\rangle\right]\alpha_{\mathbf{x}'}^* \\
&+ \int_{\mathbf{x}'} U_{BB}(\mathbf{x} - \mathbf{x}')\tilde{\varphi}_{\mathbf{x}} \, \alpha_{\mathbf{x}'}^* \alpha_{\mathbf{x}'} + \int_{\mathbf{x}'} U_{BB}(\mathbf{x} - \mathbf{x}')\tilde{\varphi}_{\mathbf{x}'} \, \alpha_{\mathbf{x}'}^* \alpha_{\mathbf{x}} \\
&+ \int_{\mathbf{x}'} U_{BB}(\mathbf{x} - \mathbf{x}')\tilde{\varphi}_{\mathbf{x}'}^* \, \alpha_{\mathbf{x}'} \alpha_{\mathbf{x}} + \int_{\mathbf{x}'} U_{BB}(\mathbf{x} - \mathbf{x}') \, \alpha_{\mathbf{x}'}^* \alpha_{\mathbf{x}'} \alpha_{\mathbf{x}} \\
&+ \int_{\mathbf{x}'} U_{BB}(\mathbf{x} - \mathbf{x}')\varphi_{rep,\mathbf{x}} \langle: \hat{\phi}_{\mathbf{x}'}^{(B)\dagger} \hat{\phi}_{\mathbf{x}'}^{(B)} :\rangle + \int_{\mathbf{x}'} U_{BB}(\mathbf{x} - \mathbf{x}')\varphi_{rep,\mathbf{x}'} \langle: \hat{\phi}_{\mathbf{x}'}^{(B)\dagger} \hat{\phi}_{\mathbf{x}}^{(B)} :\rangle \\
&+ \int_{\mathbf{x}'} U_{BB}(\mathbf{x} - \mathbf{x}')\varphi_{rep,\mathbf{x}'}^* \langle: \hat{\phi}_{\mathbf{x}}^{(B)} \hat{\phi}_{\mathbf{x}'}^{(B)} :\rangle + \int_{\mathbf{x}'} U_{BB}(\mathbf{x} - \mathbf{x}') \langle: \hat{\phi}_{\mathbf{x}'}^{(B)\dagger} \hat{\phi}_{\mathbf{x}'}^{(B)} \hat{\phi}_{\mathbf{x}}^{(B)} :\rangle \\
&+ \left[h_0 + \int_{\mathbf{x}'} U_{BB}(\mathbf{x} - \mathbf{x}')|\varphi_{rep,\mathbf{x}'}|^2\right]\langle\hat{\phi}_{\mathbf{x}'}^{(B)}\rangle + \int_{\mathbf{x}'} U_{BB}(\mathbf{x} - \mathbf{x}') \, \varphi_{rep,\mathbf{x}}^* \varphi_{rep,\mathbf{x}'} \langle\hat{\phi}_{\mathbf{x}'}^{(B)}\rangle \\
&+ \int_{\mathbf{x}'} U_{BB}(\mathbf{x} - \mathbf{x}') \, \varphi_{rep,\mathbf{x}'} \varphi_{rep,\mathbf{x}} \langle\hat{\phi}_{\mathbf{x}'}^{(B)\dagger}\rangle - \lambda u_{B,\mathbf{x}} + \lambda^* v_{B,\mathbf{x}} = 0 \,,
\end{aligned}$$
$$\text{(C1)}$$

where $\tilde{\varphi}_{\mathbf{x}} = \varphi_{rep,\mathbf{x}} + \langle\hat{\phi}_{\mathbf{x}}^{(B)}\rangle$, $\Delta\langle: \hat{\phi}_{\mathbf{x}}^{(B)(\dagger)} \hat{\phi}_{\mathbf{y}}^{(B)} :\rangle = \langle: \hat{\phi}_{\mathbf{x}}^{(B)(\dagger)} \hat{\phi}_{\mathbf{y}}^{(B)} :\rangle - \langle\hat{\phi}_{\mathbf{x}}^{(B)(\dagger)}\rangle\langle\hat{\phi}_{\mathbf{y}}^{(B)}\rangle$, and the expectation value $\langle\cdots\rangle$ is taken over $|\psi_{(B)}\rangle$. The states $|\psi_{(B)}\rangle$, respectively, the energies of the metastbale states are the eigenstates, respectively, eigen energies of

$$\hat{H}_{eff,B} = \sum_{n,m=0} \hat{H}_{n,m}[\alpha_{\mathbf{x}}^*, \alpha_{\mathbf{x}}] \, \hat{b}^{\dagger n} \hat{b}^m \,, \qquad \text{(C2)}$$

in the Fock space of $\hat{b}$, determined by exact diagonalization. The explicit eigenvalue problem is

$$\sum_l \sum_{n,m=0}^{2} \hat{H}_{n,m}[\alpha_{\mathbf{x}}^*, \alpha_{\mathbf{x}}] \, \langle k| \hat{b}^{\dagger n} \hat{b}^m |l\rangle \, \psi_l = E \, \psi_k \,, \qquad \text{(C3)}$$

where $E$ is the energy of the many-body bound state $|\psi_{(B)}\rangle = \sum_n \psi_n |n\rangle_B$.

By applying the assumptions and approximations we made in this work, the equation C1 satisfied by $\alpha_{\mathbf{x}}$ reduces to the following simplified equation

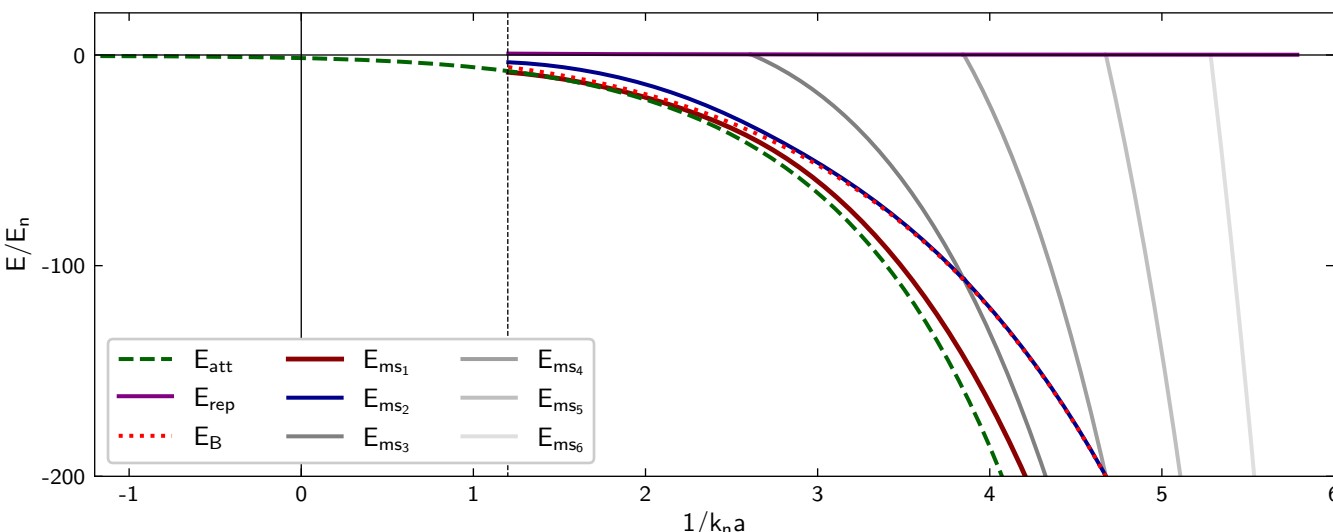

FIG. 9. Energy of polaron states across an impurity-boson Feshbach resonance, as in Fig. 2, on a linear scale (see Appendix D). The large separation of the bound state energy from the BEC energy scale is evident.

$$
\begin{aligned}
&\left[h_0 + 3U_0\tilde{\varphi}_{\mathbf{x}}^2 + 2U_0\eta_{\mathbf{x}}^2\Delta\langle\hat{b}^\dagger\hat{b}\rangle + \eta_{\mathbf{x}}^2\Delta\langle\hat{b}^2\rangle\right]\alpha_{\mathbf{x}} \\
&+ 3U_0\,\tilde{\varphi}_{\mathbf{x}}\,\alpha_{\mathbf{x}}^2 + U_0\,\alpha_{\mathbf{x}}^3 + 2\,U_0\varphi_{\mathrm{rep},\mathbf{x}}\,\eta_{\mathbf{x}}^2\langle\hat{b}^\dagger\hat{b}\rangle \\
&+ U_0\varphi_{\mathrm{rep},\mathbf{x}}\,\eta_{\mathbf{x}}^2\langle\hat{b}^2\rangle + U_0\,\eta_{\mathbf{x}}^3\langle\hat{b}^\dagger\hat{b}^2\rangle \\
&+ \left[h_0 + 2\,U_0\varphi_{\mathrm{rep},\mathbf{x}}^2\langle\hat{b}\rangle + U_0\,\varphi_{\mathrm{rep},\mathbf{x}}^2\langle\hat{b}^\dagger\rangle\right]\eta_{\mathbf{x}} - \lambda\eta_{\mathbf{x}} = 0\,.
\end{aligned}
$$
$$(C4)$$

In Eq. C4, we made use of the fact that $\alpha_{\mathbf{x}}$ can be taken

to be real, $\alpha_{\mathbf{x}} = \alpha_{\mathbf{x}}^*$.

**Appendix D: Energy of polaron states on linear scale**

In this appendix, we plot the energies of different polaron states in linear scale in Fig. 9, where it is more transparent to compare to experimental results.