# Peer review of "A unified theory of strong coupling Bose polarons: From repulsive polarons to non-Gaussian many-body bound states"

_SciPost Physics_

## Round 3 · Referee Report · Anonymous (Referee 1) · 2023-8-31

Strengths

  1. The work is timely -- there is currently a strong interest in Bose gases with impurities
  2. The work suggests a novel (to the best of my knowledge) variational approach to the problem

Weaknesses

  1. It seems that the work does not provide a rigorous validation of the proposed variational approach.

Report

The manuscript suggests a novel variational ansatz to tackle the Bose polaron problem in ultracold gases with positive scattering lengths. Using this ansatz, the paper studies attractive and repulsive quasiparticle branches. The focus is in particular on what happens to the Bose polaron spectrum across an impurity-boson Feshbach resonance, see Fig. 1 for an illustration.

I read the work with great interest. The manuscript is somewhat technical and sometimes hard to follow but overall it is written well, and will definitely be useful for researchers working on Bose polarons.

The main weakness of the paper is that it does not provide a mathematical or numerical proof of the proposed variational ansatz. This ansatz seems definitely reasonable for a
bound state with (approximately) one boson bound to an impurity, but I am not sure that I understand why it should work for N-boson-plus-impurity bound states, here N is (approximately) the number of bound bosons.

Looking at the acceptance criteria of SciPost Physics, the manuscript might "Open a new pathway in an existing or a new research direction, with clear potential for multipronged follow-up work". However, in my opinion, to meet this criteria, some further work is needed to validate the ansatz.

Requested changes

Major:

  1. The manuscript should provide a stronger justification for the use of the variational ansatz. Is it possible to estimate the effect of the neglected piece of the Hilbert space in the spirit of the mentioned in the manuscript Born-Oppenheimer approximation? Alternatively, are there any numerical results in the literature for few- or many-body systems that can be used for benchmarking?

Minor:

A. Annals of Physics 19, 234 (1962) and J. Phys. B: At. Mol. Opt. Phys. 53 205302 (2020) are relevant references that might be considered together with the mentioned mean-field studies of the Bose polaron, e.g., [61-63].

B. The manuscript states that "the repuslive [typo!] polaron cannot exist without its attractive counterpart." This statement should probably be clarified, as the repulsive polaron is expected to be a stable ground state for purely repulsive interactions, see for example Fig. 7 of Atoms 10, 55 (2022).

C. What is meant by "the third solution" on page 5 (at the very top of the right column)?

D. The manuscript uses a delta function to model boson-boson interactions. It is unclear if this is justified for the present beyond-mean-field study, i.e., when H_3 and H_4 are included. The manuscript should clarify this issue.

E. The manuscript states that "it is essential to include the effects of quantum fluctuations through Γ" for low spatial dimensions. It is worth clarifying this point. Naively, one would assume that quantum fluctuations become important only when one considers long-range physics, but it might be that I am missing something.

F. Related to E. Note that a low dimensional geometry might provide a testbed for the employed variational approach -- there are a number of various numerical techniques that can provide accurate results. Therefore, it might make sense to motivate further studies of low-dimensional Bose polarons using the proposed variational ansatz.

G. The manuscript introduces r_0 to model the range of the boson-impurity interaction. Unfortunately, the reader is left wondering what is the role of r_0 on the reported results. What will happen if r_0 is changed?

H. The manuscript states "it takes into account the quantum correlations of bound Bogoliubov excitations exactly, without restricting the number of excitations." It is worth clarifying what is meant by `exactly' here.

  • validity: good
  • significance: high
  • originality: high
  • clarity: good
  • formatting: good
  • grammar: good

Author:  Nader Mostaan  on 2024-06-13  [id 4567]

(in reply to Report 1 on 2023-08-31)

Please find the reply to the referee report in the attached file entitled "StrongCouplingBosePolaron_RefereeReply.pdf".

Attachment:

StrongCouplingBosePolaron_RefereeReply_NjAIonS.pdf

---

## Editorial Decision

resubmitted